# Sparse deep predictive coding captures contour integration capabilities of the early visual system

**Victor Boutin** [1,2] *, **Angelo Franciosini** [1], **Frederic Chavane** [1], **Franck Ruffier** [2], **Laurent Perrinet** [1]

**1** Aix Marseille Univ, CNRS, INT, Inst Neurosci Timone, Marseille, France, **2** Aix Marseille Univ, CNRS, ISM, Marseille, France

* boutin.victor@gmail.com

**Data Availability Statement:** The authors confirm that all data underlying the findings are fully available without restriction. All codes in python necessary to reproduce all the figures of the paper are released in GitHub (https://github.com/

## Abstract

Both neurophysiological and psychophysical experiments have pointed out the crucial role of recurrent and feedback connections to process context-dependent information in the early visual cortex. While numerous models have accounted for feedback effects at either neural or representational level, none of them were able to bind those two levels of analysis. Is it possible to describe feedback effects at both levels using the same model? We answer this question by combining Predictive Coding (PC) and Sparse Coding (SC) into a hierarchical and convolutional framework applied to realistic problems. In the Sparse Deep Predictive Coding (SDPC) model, the SC component models the internal recurrent processing within each layer, and the PC component describes the interactions between layers using feedforward and feedback connections. Here, we train a 2-layered SDPC on two different databases of images, and we interpret it as a model of the early visual system (V1 & V2). We first demonstrate that once the training has converged, SDPC exhibits oriented and localized receptive fields in V1 and more complex features in V2. Second, we analyze the effects of feedback on the neural organization beyond the classical receptive field of V1 neurons using interaction maps. These maps are similar to association fields and reflect the Gestalt principle of good continuation. We demonstrate that feedback signals reorganize interaction maps and modulate neural activity to promote contour integration. Third, we demonstrate at the representational level that the SDPC feedback connections are able to overcome noise in input images. Therefore, the SDPC captures the association field principle at the neural level which results in a better reconstruction of blurred images at the representational level.

## Author summary

One often compares biological vision to a camera-like system where an image would be processed according to a sequence of successive transformations. In particular, this "feed-forward" view is prevalent in models of visual processing such as deep learning. However, neuroscientists have long stressed that more complex information flow is necessary to reach natural vision efficiency. In particular, recurrent and feedback connections in the

VictorBoutin/InteractionMap). The two databases used to train the algorithm are publicly available online in the following link: STL-10 database (http://cs.stanford.edu/~acoates/stl10) and Chicago Face Database (https://chicagofaces.org/default/).

**Funding:** VB and LP received funding from the European Union's H2020 research and innovation programme under the Marie Sklodowska-Curie grant agreement n 713750, by the Regional Council of Provence-Alpes-Côte d'Azur, A*MIDEX (n ANR-11-IDEX-0001-02), and the financial support of ANR project "Horizontal-V1" (n ANR-17-CE37-0006). This work was granted access to the HPC resources of Aix-Marseille Université financed by the project Equip@Meso (n ANR-10-EQPX-29-01) of the program "Investissements d'Avenir". Other authors received no specific funding for this work. The funders had no role in study design, data collection and analysis, decision to publish, or preparation of the manuscript.

**Competing interests:** The authors have declared that no competing interests exist.

visual cortex allow to integrate contextual information in our representation of visual stimuli. These modulations have been observed both at the low-level of neural activity and at the higher level of perception. In this study, we present an architecture that describes biological vision at both levels of analysis. It suggests that the brain uses feedforward and feedback connections to compare the sensory stimulus with its own internal representation. In contrast to classical deep learning approaches, we show that our model learns interpretable features. Moreover, we demonstrate that feedback signals modulate neural activity to promote good continuity of contours. Finally, the same model can disambiguate images corrupted by noise. To the best of our knowledge, this is the first time that the same model describes the effect of recurrent and feedback modulations at both neural and representational levels.

## Introduction

Visual processing of objects and textures has been traditionally described as a pure feedforward process that extracts local features. These features become increasingly more complex and task-specific along the hierarchy of the ventral visual pathway [1, 2]. This view is supported by the very short latency of evoked activity observed in monkeys ($\approx 90$ ms) in higher-order visual areas [3, 4]. The feed-forward flow of information is sufficient to account for core object categorization in the IT cortical area [5]. Although this feedforward view of the visual cortex was able to account for a large scope of electrophysiological [6, 7] and psychophysical [8] findings, it does not take advantage of the high density ($\approx 20\%$) and diversity of feedback connections observed in the anatomy [9–11].

Feedback connections, but also horizontal intra-cortical connections are known to integrate contextual modulations in the early visual cortex [12–14]. At the neurophysiological level, it was observed that the activity in the center of the Receptive Field (RF), called the classical RF, was either suppressed or facilitated by neural activity in the surrounding regions (i.e. the extra-classical RFs). These so-called 'Center/Surround' modulations are known to be highly stimulus specific [15]. For example, when gratings are presented to the visual system, feedback signals tend to suppress horizontal connectivity which is thought to better segregate the shape of the perceived object from the ground (figure-ground segregation) [16, 17]. In contrast, when co-linear and co-oriented lines are presented, feedback signals facilitate horizontal connections such that local edges are grouped towards better shape coherence (contour integration) [18]. Interestingly, both figure-ground segregation and contour integration are directly derived from the Gestalt principle of perception. In particular, contour integration is known to follow the Gestalt rule of good continuation as mathematically formalized by the concept of association field [19]. This association field suggests that local edges tend to align toward a co-circular/co-linear geometry. Besides being central in natural image organization [20], association fields might also be implemented in the connectivity within the V1 area [21, 22] and play a crucial role in contour perception [19, 23]. In particular, it was demonstrated that short-range feedback connections (originating in the ventral visual area and targeting V1) play a crucial role in the recognition of degraded images [24]. These pieces of biological evidence suggest that feedforward models are not sufficient to account for the context-dependent behavior of the early visual cortex and urge us to look for more complex models taking advantage of recurrent connections.

From a computational perspective, both Predictive Coding (PC) and Sparse Coding (SC) are good candidates to model the early visual system. On one hand, SC might be considered as

a framework to describe local computations in the early visual cortex. Olshausen & Field demonstrated that a SC strategy was sufficient to account for the emergence of features similar to the Receptive Fields (RFs) of simple cells in the mammalian primary visual cortex [25]. These RFs are spatially localized, oriented band-pass filters [26]. Furthermore, SC could also be considered to result from a competitive mechanism. SC implements an "explaining away" strategy [27] by selecting only the dominant causes of the sensory input. On the other hand, PC describes the brain as a Bayesian process that consistently updates its internal model of the world to infer the possible physical causes of a given sensory input [28, 29]. PC suggests that top-down connections convey predictions about the activity in the lower level while bottom-up processes transmit prediction error to the higher level. In particular, PC models were able to describe center-surround antagonism in the retina [30] and extra-classical RFs effects observed in the early visual cortex [29]. In addition, studies have investigated the correspondence between cortical micro-circuitry and the connectivity implied by the PC theory [31, 32]. Therefore, while SC might be considered as a local mechanism modeling recurrent computation within brain areas, PC leverages top-down connections to describe interactions between cortical areas at a more global scale.

Rao & Ballard [29] were the first to leverage Predictive Coding (PC) into a hierarchical framework and to combine it with Sparse Coding (SC). The 2-layers PC model they have proposed had few dozens of neurons (20 in the first layer and 32 in the second one) linked with fully connected synapses and trained on patches extracted from 5 natural images. These settings did not allow the authors to spatially extend their analysis to the effect of the feedback outside of the classical RF and to train their network on a scale that is more realistic (i.e. higher resolution images and more neurons). In contrast, recently proposed architectures in deep learning, like autoencoders, allow to successfully tackle larger-scale problems. Both PC and autoencoders describe the generative process that gives rise to a given observation through a Bayes decomposition of a probabilistic model and using a hierarchy of latent representation [33]. Both frameworks can also be regularized using a sparse constraint on the latent representation (see [34, 35] for more details on sparse autoencoders). Nevertheless, PC and autoencoders are exhibiting 3 major differences. First, while the encoder/decoder are different in autoencoders, these are tied in PC. Second, autoencoders reconstruct the input image whereas a PC layer aims at reconstructing the previous layer latent variables (i.e. only the first layer aims at reconstructing the input image in PC). Third, autoencoders are mostly trained by back-propagation to minimize a unique global reconstruction error while PC is trained to jointly minimize several local reconstruction errors. Interestingly, other convolutional PC frameworks, formulated to solve discriminative problems, have recently emerged to propose a local approximation of the back-propagation algorithms in the domain of classification tasks [36].

In this paper, we use a Sparse Deep Predictive Coding (SDPC) model that combines Predictive Coding and Sparse Coding into a convolutional neural network. The proposed model leverages the latest technics used in deep learning to extend the original PC framework [29] to larger scale (higher resolution images seen by hundreds of thousands of neurons). While the Rao & Ballard PC model describes the contextual effects of the feedback connection in the classical RFs, we leverage the convolutional structure of our network over a larger scale to address the question of these contextual influences outside of the classical RF (i.e. in the extra-classical RF). The main novelty of the SDPC lies in 3 main aspects. First, the SDPC is extending to larger scale the original PC framework while keeping a learning approach that relies on the minimization of local reconstruction errors that could be interpreted as Hebbian learning (as opposed to autoencoders that minimize a global loss function). Second, it includes the latest Sparse Coding (SC) technics to constrain the latent variables (i.e. iterative soft-thresholding

algorithms). Third, the convolutional approach adopted in the SDPC allows to extend the analysis made by Rao & Ballard to neurons located in the extra-classical RFs.

We first briefly introduce the 2-layered SDPC network used to conduct all the experiments of the paper, and we show the results of the training of the SDPC on two different databases. Next, we investigate the feedback effects at the "neural" level. We show how feedback signals in SDPC account for a reshaping of V1 neural population both in terms of topographic organization and activity level. Then, we probe the effect of feedback at the "representational" level. In particular, we investigate the ability of feedback connections to denoise input images. Finally, we discuss the results obtained with the SDPC model in the light of the psychophysical and neurophysiological findings observed in neuroscience.

## Results

In our mathematical description of the proposed model, italic letters are used as symbols for *scalars*, bold lowercase letters for column **vectors** and bold uppercase letters for **MATRICES**. $j$ refers to the complex number such that $j^2 = -1$.

### Brief description of the SDPC

Given a hierarchical generative model for the formation of images, the core objective of hierarchical Sparse Coding (SC) is to retrieve the parameters and the internal states variables (i.e. latent variables) that best explain the input stimulus. As any perceptual inference model, hierarchical SC attempts to solve an inverse problem (Eq 1), where the forward model is a hierarchical linear model [37]:

$$\begin{cases} \boldsymbol{x} = \mathbf{D}_1^T \boldsymbol{\gamma}_1 + \epsilon_1 & \text{s.t.} \quad \|\boldsymbol{\gamma}_1\|_0 < \alpha_1 \quad \text{and} \quad \boldsymbol{\gamma}_1 > 0 \\ \boldsymbol{\gamma}_1 = \mathbf{D}_2^T \boldsymbol{\gamma}_2 + \epsilon_2 & \text{s.t.} \quad \|\boldsymbol{\gamma}_2\|_0 < \alpha_2 \quad \text{and} \quad \boldsymbol{\gamma}_2 > 0 \\ .. \\ \boldsymbol{\gamma}_{L-1} = \mathbf{D}_L^T \boldsymbol{\gamma}_L + \epsilon_L & \text{s.t.} \quad \|\boldsymbol{\gamma}_L\|_0 < \alpha_L \quad \text{and} \quad \boldsymbol{\gamma}_L > 0 \end{cases} \quad (1)$$

The number of layers of our model is denoted $L$ and $\boldsymbol{x}$ is the sensory input (i.e. image). The sparsity at each layer is enforced by a constraint on the $\ell_0$ pseudo-norm of the internal state variable $\boldsymbol{\gamma}_i$. Note that this operator is termed pseudo-norm as it is counting the number of strictly positive scalars and does not depend on their amplitude. Finally, $\boldsymbol{\epsilon}_i$ and $\mathbf{D}_i$ are respectively the prediction error (i.e. reconstruction error) and the weights (i.e. the parameters) at each layer $i$.

To tighten the link with neuroscience, we impose $\boldsymbol{\gamma}_i$ to be non-negative such that the element of the internal state variables could be interpreted as firing rates. In addition, we include convolutional synaptic weights, as the underlying weight sharing mechanism is well modeling the position invariance of features within natural images. It allows us to interpret $\boldsymbol{\gamma}_i$ as a retinotopic map describing the neural activity at layer $i$, and we call it the activity map. Mathematically speaking, our activity maps are 3D tensors. An activity map, $\boldsymbol{\gamma}_i$ of size $[n_f, w_m, h_m]$ could be interpreted as a collection of $n_f$ 2D maps of dimension $(w_m, h_m)$. In a convolutional setting, $\mathbf{D}_i$ of size $[n_f, n_c, w, h]$ could be viewed as a collection of $n_f$ features of size $n_c \times w \times h$. The width and height of the features are denoted by $w$ and $h$, respectively. $\mathbf{D}_i$ is called a dictionary. In terms of neuroscience, $\mathbf{D}_i$ could be viewed as the synaptic weights between 2 layers whose activity is represented by $\boldsymbol{\gamma}_{i-1}$ and $\boldsymbol{\gamma}_i$. In this article all the matrix-vector products correspond to discrete 2D spatial convolutions (see Eq 2 for the mathematical definition of the discrete 2D convolution). To facilitate the reading of the mathematical equation, we have purposely abused

the notations by replacing all the 2D convolutions operator by matrix-vector products (in 'Model and methods' we explain the mathematical equivalence between convolutions and matrix-vector products).

$$\boldsymbol{\gamma}_{i-1} = \mathbf{D}_i^T * \boldsymbol{\gamma}_i$$

$$\text{with } \boldsymbol{\gamma}_{i-1}[j,k,l] = \sum_{m=1}^{n_c}\sum_{p=1}^{w}\sum_{q=1}^{h}\mathbf{D}_i^T[j,m,p,q] \times \boldsymbol{\gamma}_{i-1}[m,k-p,l-q] \tag{2}$$

$$\text{s.t. } k-p \in [\![1,w_m]\!] \text{ and } l-q \in [\![1,h_m]\!]$$

In Eq 3, we define the effective dictionary, $\mathbf{D}_i^{\mathrm{eff}}$, as the back-projection of $\mathbf{D}_i$ into the visual space (see S7 Fig for an illustration).

$$\mathbf{D}_i^{\mathrm{eff}^T} = \mathbf{D}_1^T..\mathbf{D}_{i-1}^T\mathbf{D}_i^T \tag{3}$$

The effective dictionaries could also be interpreted as a set of Receptive Fields (RFs). Note that RFs in the visual space get bigger for neurons located in deeper layers (that is, on layers further away from the sensory layer). To visualize the information represented by each layer, we back-project $\gamma_i$ into the visual space (see Eq 18). We call this projection a "representation" and it is denoted by $\gamma_i^{\mathrm{eff}}$.

One possibility to solve the problem defined by Eq 1 in a neuro-plausible way is to use the Sparse Deep Predictive Coding (SDPC) model. The SDPC model combines local computational mechanisms to learn the weights and infer internal state variables. It leverages recurrent and bi-directional connections (feedback and feedforward) through the Predictive Coding (PC) theory. In this paper, we aim at modeling the early visual cortex using a 2-layered version of the SDPC. Consequently, we denote the first and second layers of the SDPC as the V1 and V2 models, respectively. Our V1 and V2 models are driven by the joint minimization of the $\mathcal{L}_1$ and $\mathcal{L}_2$ loss function (see Eq 4). In the section 'Model and methods', the Eq 15 described a generalized version of the loss function.

$$\begin{cases} \mathcal{L}_1 = & \dfrac{1}{2}\parallel \boldsymbol{x} - \mathbf{D}_1^T\boldsymbol{\gamma}_1 \parallel_2^2 + \dfrac{k_{\mathrm{FB}}}{2}\parallel \boldsymbol{\gamma}_1 - \mathbf{D}_2^T\boldsymbol{\gamma}_2 \parallel_2^2 + \lambda_1\parallel \boldsymbol{\gamma}_1 \parallel_1 \\[4mm] \mathcal{L}_2 = & \dfrac{1}{2}\parallel \boldsymbol{\gamma}_1 - \mathbf{D}_2^T\boldsymbol{\gamma}_2 \parallel_2^2 + \lambda_2\parallel \boldsymbol{\gamma}_2 \parallel_1 \end{cases} \tag{4}$$

In Eq 4, $\lambda_i$ is a scalar that controls the sparsity level within each layer. Note that we have relaxed the $\ell_0$-norm regularization in Eq 1 by replacing it with a $\ell_1$-norm constraint in Eq 4. The parameter $k_{\mathrm{FB}}$ is used to increase the strength of the representation error coming from V2. We thus call $k_{\mathrm{FB}}$ 'the feedback strength' as it allows us to tune how close the V1 neural activity is from its prediction made by V2. Last but not least, when the parameter $k_{\mathrm{FB}}$ is set to 0, the SDPC becomes a stacking of independent LASSO sub-problems [38, 39] and is not relying anymore on the Predictive Coding (PC) framework. Consequently, we also use the $k_{\mathrm{FB}}$ parameter to evaluate the effect of the PC on the first layer representation. At a first glance, it seems to be sub-optimal to use $k_{\mathrm{FB}} \neq 1$ in the loss function defined in Eq 4 to solve Eq 1. However we will see in the rest of the manuscript, that higher feedback strength provide the SDPC with several advantages both in terms of neuroscience interpretation (see section 'Effect of the feedback at the neural level') and in terms of computation (see section 'Effect of the feedback at the representational level'). In Eq 4, $\gamma_1$ corresponds to the activity-map in V1 and $\gamma_2$ to V2's

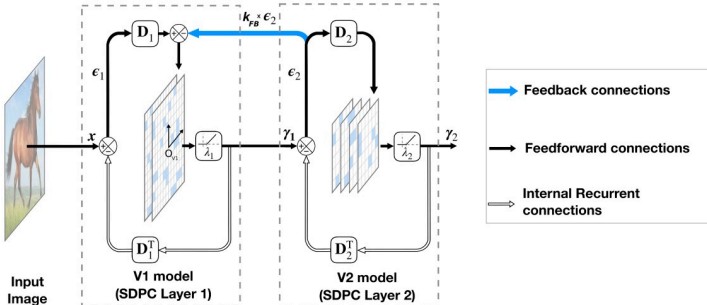

**Fig 1. Architecture of a 2-layered SDPC model.** In this model, $\gamma_i$ represents the activity of the neural population and $\epsilon_i$ is the representation error (also called prediction error) at layer $i$. The synaptic weights of the feedback and feedforward connection at each layer ($\mathbf{D}_i^T$ and $\mathbf{D}_i$ respectively) are reciprocal. The level of sparseness is tuned with the soft-thresholding parameter $\lambda_i$. The scalar $k_{FB}$ controls the strength of the feedback connection represented with a blue arrow.

activity-map. We refer to the V1 space as the retinotopic space described by $\gamma_1$, and it is symbolized with a small coordinate system centered in $O_{V1}$ (see Fig 1).

The joint optimization of the loss function described in Eq 4 is performed using an alternation of inference and learning step. The inference step involves finding the activities (i.e. $\gamma_i$) that minimize $\mathcal{L}_i$ using Eq 5. In this equation, the sparsity constraint is achieved using a soft-thresholding operator, denoted $\mathcal{T}_\alpha(\cdot)$ (see Eq 16 in section 'Model and methods' for the mathematical definition of the soft-thresholding operator).

$$
\begin{aligned}
\gamma_i^{t+1} &= \mathcal{T}_{\eta_{c_i}\lambda_i}\left(\gamma_i^t - \eta_{c_i}\frac{\partial \mathcal{L}_i}{\partial \gamma_i^t}\right) \\
&= \mathcal{T}_{\eta_{c_i}\lambda_i}\left(\gamma_i^t + \eta_{c_i}\mathbf{D}_i(\gamma_{i-1}^t - \mathbf{D}_i^T\gamma_i^t) - k_{FB} \cdot \eta_{c_i}(\gamma_i^t - \mathbf{D}_{i+1}^T\gamma_{i+1}^t)\right)
\end{aligned}
\tag{5}
$$

Once the inference procedure has reached a fixed point (see Eq 17 in section 'Model and methods' for more details on the criterion we use to define a fixed point), the SDPC learns the synaptic weight using Eq 6.

$$
\begin{aligned}
\mathbf{D}_i^{t+1} &= \mathbf{D}_i^t - \eta_{L_i}\frac{\partial \mathcal{L}_i}{\partial \mathbf{D}_i} \\
&= \mathbf{D}_i^t + \eta_{L_i}\gamma_i^T(\gamma_{i-1} - \mathbf{D}_i^t T\gamma_i)
\end{aligned}
\tag{6}
$$

In both Eqs 5 and 6, the variables $\gamma_i^t$ and $\mathbf{D}_i^t$ denote the neural activity and the synaptic weight at time step t, respectively. $\eta_{c_i}$ is defining the time step of the inference process and $\eta_{L_i}$ is the learning rate of the learning process. We train the SDPC on 2 different datasets: a face database and a natural images database.

In this paper, we aim at modeling the early visual cortex using a 2-layered version of SDPC (see Fig 1). Consequently, we denote the first and second layer of the SDPC as the V1 and V2 model, respectively. All presented results are obtained with a SDPC network trained with a feedback strength equal to 1 (i.e $k_{FB} = 1$). Once trained, and when specified, we vary the feedback strength to evaluate its effect on the inference process. Note that we have also experimented to equate the feedback strength during learning and inference, and the results obtained are extremely similar to those obtained when the feedback strength was set to 1 during the SDPC training. For both databases, all the presented results are obtained on a testing set that is different from the training set (except when we describe the training in the section

entitled 'SDPC learns localized edge-like RFs in V1 and more specific RFs in V2'). All network parameters and database specifications are listed in the 'Model and methods' section.

## SDPC learns localized edge-like RFs in V1 and more specific RFs in V2

In this subsection, we present the results of the training of the Sparse Deep Predictive Coding (SDPC) model on both the natural images and the face databases with a feedback strength $k_{FB}$ equal to 1 (Fig 2). First-layer Receptive Fields (RFs) exhibit two different types of filters: low-frequency filters, and higher frequency filters that are localized, band-pass and similar to Gabor filters (Fig 2B and 2F). The low-frequency filters are mainly encoding for textures and colors whereas the higher frequency ones describe contours. Second layer RFs (Fig 2D and 2G) are built from a linear combination of the first layer RFs. For both databases, the second layer RFs are bigger than those in the first layer (approximately 3 times bigger for both databases). We note that for the face database the second layer RFs present curvatures and specific face features, whereas on the natural images database they only exhibit longer oriented edges. This difference is mainly coming from the higher variety of natural images: the identity of

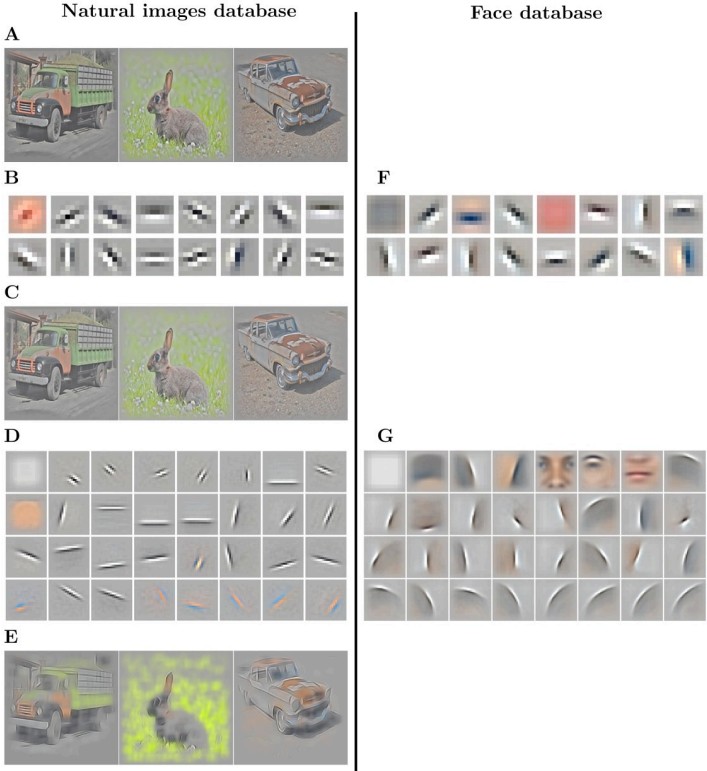

**Fig 2. Results of training SDPC on the natural images (left column) and on the face database (right column) with a feedback strength $k_{FB}$ = 1.** (**A**): Randomly selected input images from the natural images database (denoted $x$ in the text). The two databases are pre-processed with Local Contrast Normalization [40] and whitening. (**B**) & (**F**): 16 randomly selected first-layer Receptive Fields (RFs) from the 64 RFs composing $D_1^{\text{eff}^T}$ (note that $D_1^{\text{eff}^T} = D_1^T$, see Eq 3). The RFs are ranked by their activation probability in a descending order. The RF size of neurons located on the first layer is 9 × 9 px for both databases. (**C**): Reconstruction of images corresponding to the input images shown in (**A**) from the representation in the first layer, denoted $\gamma_1^{\text{eff}}$ (note that $\gamma_1^{\text{eff}} = D_1^T \gamma_1$, see Eq 18). (**D**) & (**G**): 32 sub-sampled RFs out of 128 RFs composing $D_2^{\text{eff}^T}$ (note that $D_2^{\text{eff}^T} = D_1^T D_2^T$, see Eq 3), ranked by their activation probability in descending order. The size of the RF from neurons located on the second layer is 22 × 22 px on the natural images database (**D**) and 33 × 33 px on the face database (**G**). (**E**): Reconstruction of images corresponding to the input images shown in (**A**) from the representation in the second layer, denoted $\gamma_2^{\text{eff}}$ (note that $\gamma_2^{\text{eff}} = D_1^T D_2^T \gamma_2$, see Eq 18).

objects, their distances, and their angles of view are more diverse than in the face database. On the contrary, as the face database is composed only of well-calibrated, centered faces, the SDPC model is able to extract curvatures and features that are common to all faces. In particular, we observe on the face database the emergence of face-specific features such as eyes, nose or mouth that are often selected by the model to describe the input (second layer RFs are ranked by their activation probability in descending order in Fig 2D and 2G). All the 64 first layer RFs and the 128 second layer RFs learned by the SDPC on both databases are available in the Supporting information section (S1 Fig for natural images and S2 Fig for face database). The first layer reconstruction (Fig 2C) is highly similar to the input image (Fig 2A). In the second layer reconstructions (Fig 2E), the details like textures and colors are faded and smoothed in favor of more pronounced contours. In particular, the contours of the natural images reconstructed by the second layer of the SDPC are sketched with a few oriented lines.

## Effect of the feedback at the neural level

We now vary the strength of the feedback connection to assess its impact on neural representations when an image is presented as a stimulus. The strength of the feedback, $k_{FB}$, is a scalar ranging from 0 to 4. When $k_{FB}$ is set to 0, the feedback connection is suppressed. In other words, the neural activity in the first layer is independent of the neural activity in the second layer. Inversely, when $k_{FB} = 4$, the feedback signals are strongly amplified such that it reinforces the interdependence between the neural activities of both layers. As a consequence, varying feedback strength should also affect the activity in the first layer. The objective of this subsection is to study the effect of the feedback on the organization of V1 neurons (i.e. the first layer of the SDPC).

**SDPC feedback recruits more neurons in the V1 model.** In the first experiment, we monitor the median number of active neurons in our V1 model when varying the feedback strength on both databases. The medians are computed over 1200 images of natural images database (Fig 3A) and 400 images of the face database (Fig 3B). In this paper, we use the median ± median absolute deviation instead of the classical mean ± standard deviation to avoid assuming that samples are normally distributed [41]. For the same reason, all the statistical tests are performed using the Wilcoxon signed-rank test. It will be denoted $WT(N = 1200, p < 0.01)$ when the null hypothesis is rejected. In this notation, $N$ is the number of samples and $p$ is the corresponding probability value (p-value). In contrast, we will formalize the test by $WT(N = 1200, p = 0.3)$ when the null hypothesis cannot be rejected.

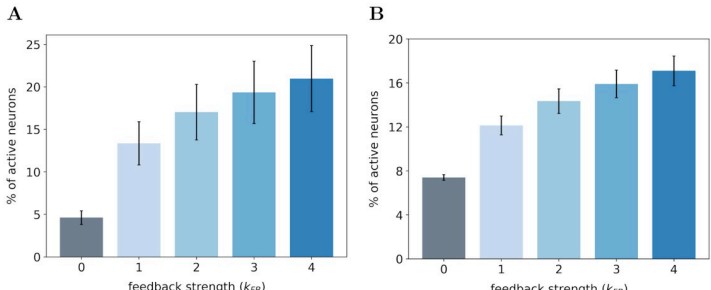

**Fig 3. Percentage of active neurons in the first layer of the SDPC model. (A)** On the natural images database. **(B)** On the face database. We record the percentage of active neurons with a feedback strength $k_{FB}$ varying from 0 (no feedback) to 4 (strong feedback). The height of the bars represent the median percentage of active neurons and the error bars are computed using the median absolute deviation over 1200 and 400 images of the testing set for the natural images and face database, respectively.

For both databases, we observe that the percentage of active neurons increases with the strength of the feedback. In particular, we note a strong increase in the number of activated neurons when we restore the feedback connection (from $k_{FB} = 0$ to $k_{FB} = 1$): +8.7% and +4.7% for natural images and face databases, respectively. Incrementally amplifying the feedback strength above 1 further increases the number of active neurons in the first layer even if the effect is sublinear. All the increases in the percentage of recruited neurons with the feedback strength are significant as quantified with a Wilcoxon signed-rank test (WT) between all pairs of feedback strength: WT($N = 1200$, $p < 0.01$) for natural images database and WT($N = 400$, $p < 0.01$) for the face database. For each database, we notice that the inter-stimuli variability, as illustrated by error-bars, is lower when the feedback connection is removed: 0.80% with $k_{FB} = 0$ versus 2.55% with $k_{FB} = 1$ for the natural images database and 0.25% with $k_{FB} = 0$ versus 0.85% with $k_{FB} = 1$ for the face database. These results lead us to 2 different conclusions: (1) The feedback connection tends to recruit more neurons in our model of V1, (2) the feedback signal is dependent of the input stimuli and leads to differentiated effect of the feedback strength.

**Interaction map to visualize the neural organization.** The V1 activity-map ($\gamma_1$) being a high-dimensional tensor, it is a priori difficult to visualize its internal organization. Following our mathematical convention, the activity maps are 3-dimensional tensors of size $[n_f, w_m, h_m]$, in which the first dimension is describing the feature space (denoted $\theta$), and the 2 last dimensions are related to spatial positions ($x$ and $y$ respectively). One could interpret the activity maps as a collection of $n_f$ 2-dimensional maps describing each feature's activity in the retinotopic space. Said differently, the scalar $\gamma_1[j, k, l]$ is quantifying how strongly correlated is the feature j (mathematically described by $\boldsymbol{D}_1^T[j, :, :, :]$) with the input image (i.e. $\boldsymbol{x}$) at the spatial location of coordinate $(k, l)$. Consequently, we denote $\theta$ the space that describes the $n_f$ features, and we call it the feature space. In practice, we extracted one angle per RF to describe its orientation by fitting the first layer features (i.e. those presented in Fig 2B and 2F) with Gabor filters [42]. Note that textural and low-frequency filters which are poorly fitted are simply filtered-out (we remove 13 out of 64 filters). We use the extracted angles to discretized the feature space: $\theta \in \{\theta_k\}_{k=0}^{n_f}$. Similarly, we describe a space of spatial coordinate (x, y) such that $x \in [\![1, w_m]\!]$ and $y \in [\![1, h_m]\!]$. One concise way to describe the V1 representation is to formalize it using the complex number notation denoted $\gamma_1^{\mathbb{C}}$ (see Eq 7).

$$\forall \theta \in \{\theta_k\}_{k=0}^{n_f}, \forall x \in [\![1, w_m]\!], \forall y \in [\![1, h_m]\!], \quad \gamma_1^{\mathbb{C}}[\theta, x, y] = \gamma_1[\theta, x, y]\, e^{j\theta}$$

$$\text{s.t. } j \in \mathbb{C} \text{ and } j^2 = -1$$

(7)

We decompose the computation of the interaction map into 3 steps (see Fig 4 for an illustration of the computation of the interaction map).

- **Step 1** is to extract small neighborhoods around the 10 most strongly activated neurons for each orientation. First, we choose a feature (i.e. an orientation, denoted $\theta_c$), and we identify the position of the 10 neurons that are the most strongly responsive to the selected orientation. Second, we extract a spatial neighborhoods of size $9 \times 9$ centered on each of these 10 neurons (we thus extract 10 different neighborhoods). We set the size of the neighborhood to be the same than the one of the V2 features (i.e. $\boldsymbol{D}_2$) so that we can capture the feedback effect coming from V2. At this stage, we have a 10 cropped versions of $\gamma_1$ which are centered on neurons strongly responsive to a given orientation. This orientation is called the central preferred orientation (still denoted $\theta_c$). We use the notation $(x_c, y_c)$ to describe the spatial coordinates of neurons belonging to the cropped version of $\gamma_1$

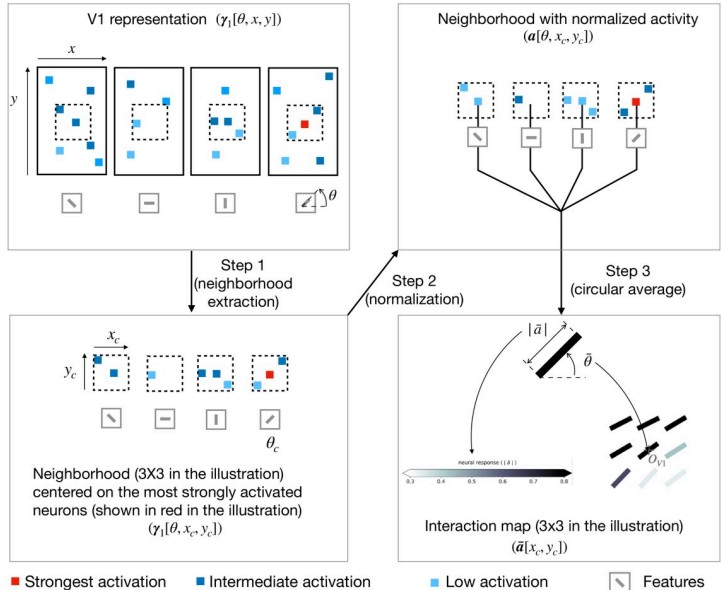

**Fig 4. Illustration of the procedure to generate interaction map.** In this illustrative example we consider a V1 representation with only 4 feature maps (represented in the upper-left box). **Step 1** is to extract a neighborhood (of size 3x3 in the illustration only) around the most strongly activated neuron (represented with a red square in the illustration) for a given central preferred orientation (denoted $\theta_c$). **Step 2** is to normalize the neural activity in the extracted neighborhood using the marginal activity (see Eq 8). **Step 3** is to compute the resulting orientation and activity at every position of the neighborhood using a circular mean (see Eqs 11 and 12 respectively). To keep a concise figure we have illustrated the computation of the central edge of the interaction map only. For simplification, the illustration shows only 1 neighborhood extraction whereas the interaction maps shown in the paper are computed by averaging neighborhoods centered on the 10 most strongly activated neurons.

- **Step 2** is to normalize the activity of the neurons in each of 10 cropped versions of $\gamma_1$ generated at **Step 1**. To normalize activity, we use the marginal activity, as defined by the mean neighborhood in a spatially shuffled version of the V1 activity-map. Said differently, the marginal activity, denoted $\gamma_1[\theta, x_{\sim c}, y_{\sim c}]$, is a spatial average over the activity of neurons that respond to one given orientation $\theta$. The variables $(x_{\sim c}, y_{\sim c})$ represent the V1-space outside of this neighborhood. We call $\boldsymbol{a}$ the normalized activity and its computation is defined in Eq 8.

$$\boldsymbol{a}[\theta, x_c, y_c] = \frac{\gamma_1[\theta, x_c, y_c] - \gamma_1[\theta, x_{\sim c}, y_{\sim c}]}{\gamma_1[\theta, x_{\sim c}, y_{\sim c}]} \tag{8}$$

Intuitively, for a given $(\theta, x_c, y_c)$, $\boldsymbol{a}[\theta, x_c, y_c]$ is positive if the activity inside the neighborhood is above the marginal activity and negative in the opposite case.

- **Step 3** is the actual computation of the interaction map. The interaction map, denoted $\bar{\boldsymbol{a}}$, is computed as the weighted average over all the orientations of the adjusted activity vector (see Eq 9). We denote $\bar{\boldsymbol{\theta}}$ and $|\bar{\boldsymbol{a}}|$ the resulting orientation and activity of the interaction map, respectively (see Eq 10).

$$\bar{\boldsymbol{a}}[x_c, y_c] \quad = \frac{1}{n}\sum_{\theta=\theta_1}^{\theta_n} \boldsymbol{a}[\theta, x_c, y_c] \cdot e^{j\theta} \tag{9}$$

$$= |\bar{\boldsymbol{a}}[x_c, y_c]| \cdot e^{j\bar{\theta}[x_c, y_c]} \tag{10}$$

We use a circular weighted average to compute the resulting orientation (see Eq 11) and activity (see Eq 12) of the interaction map.

$$\bar{\boldsymbol{\theta}}[x_c, y_c] \quad = atan2\left(\frac{1}{n}\sum_{\theta=\theta_1}^{\theta_n}\boldsymbol{a}[\theta, x_c, y_c]sin(\theta), \frac{1}{n}\sum_{\theta=\theta_1}^{\theta_n}\boldsymbol{a}[\theta, x_c, y_c]cos(\theta)\right) \quad (11)$$

$$|\bar{\boldsymbol{a}}[x_c, y_c]| \quad = \frac{1}{n}\sqrt{\left(\sum_{\theta=\theta_1}^{\theta_n}\boldsymbol{a}[\theta, x_c, y_c]cos(\theta)\right)^2 + \left(\sum_{\theta=\theta_1}^{\theta_n}\boldsymbol{a}[\theta, x_c, y_c]sin(\theta)\right)^2} \quad (12)$$

The *atan*2 operator in Eq 11 denotes a generalization of the arctangent operator that returns positive angle for counterclockwise angle and opposite for clockwise angle.

At the end of **Step 3**, we then have generated 10 interaction maps (one for each of the 10 most strongly activated neurons) for a given central preferred orientation. Next, we iterate from **Step 1** to **Step 3**, over 1200 natural images and we average the corresponding interaction maps. At this point, we then have an interaction map for one given central preferred orientation. We then repeat this process for all central preferred orientations and for different feedback strengths ranging from 0 to 4.

**SDPC feedback signals reorganize the interaction map of the V1 model.** We investigate the effect of feedback on the neural organization in our V1 model when the Sparse Deep Predictive Coding (SDPC) is trained on natural images. To conduct such an analysis, we used the concept of interaction map, as introduced previously.

For all feedback strengths and different central preferred orientations, we observe that the interaction maps are highly similar to association fields [19]: most of the orientations of the interaction map are co-linear and/or co-circular to the central preferred orientation (see Fig 5 for one example of this phenomenon and S3 Fig for more examples with $k_{FB} = 1$). In addition, interaction maps exhibit a strong activity in the center and towards the end-zone of the central preferred orientation. We define the end-zone as the region covering the axis of the central

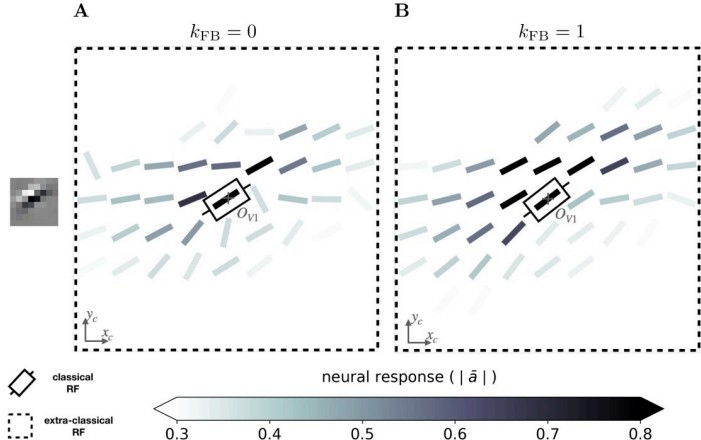

**Fig 5. Example of a $9 \times 9$ interaction map of a V1 area centered on neurons strongly responding to a central preferred orientation of 30˚. (A)** Without feedback. **(B)** With a feedback strength equal to 1. These interaction maps are obtained when the SDPC is trained on natural images. At each location identified by the coordinates $(x_c, y_c)$ the angle is $\bar{\theta}[x_c, y_c]$ (see Eq 11) and the color scale is $|\bar{a}[x_c, y_c]|$ (see Eq 12). The color scale being saturated toward both maximum and minimum activity, all the activities above 0.8 or below 0.3 have the same color, respectively dark or white.

preferred orientation, and the side-zone as the area covering the orthogonal axis of the central preferred orientation. The activity of the interaction map in the side-zone is lower compared to the activity in the end-zone. We notice qualitatively that the orientations of the interaction maps are less co-linear to the central preferred orientation when feedback is suppressed (i.e. $k_{FB} = 0$). In other words, when feedback is active, the interaction map looks more organized compared to the interaction map generated without feedback (see Fig 5 for a striking example of this phenomenon).

We next quantify this organizational difference we observed when we turn-on the feedback connection. For a given feedback strength $k_{FB}$, we introduce two ratios to assess the change of the co-linearity ($r_{\theta_{co-lin}}(k_{FB})$) and co-circularity ($r_{\theta_{co-cir}}(k_{FB})$) w.r.t. to their respective measure without feedback (see Eqs 23 and 24 in section Model and methods for mathematical details).

We report these two ratios for the end-zone (Fig 6A) and the side-zone (Fig 6B). For all $k_{FB}$ > 0, we observe that neurons located in the end-zone and in the side-zone are more co-linear to the central preferred orientation when the feedback connection is turned on. Indeed, all the bars in the left-block of Fig 6A and 6B are always above the baseline (as computed as the co-linearity / co-circularity when the feedback is turned off). This increase of co-linearity w.r.t. to the baseline is highly significant as measured with the Wilcoxon signed-rank test (WT($N = 51$, $p < 1e - 3$)). We also observe that the increase of co-linearity is more pronounced in the side-zone (co-linearity bars in Fig 6A exhibit lower values than those in Fig 6B). In addition, we note that increasing the feedback strength has a significant effect on the co-linearity in the side-zone as quantified by all pair-wise statistical tests (WT($N = 51$, $p < 1e - 2$)). In contrast, increasing the feedback strength has no effect on the co-linearity in the end-zone. We observe that the feedback is not changing the co-circularity for neurons located in both the end-zone and the side-zone. Indeed, all the bars in the right-block of Fig 6A and 6B are near the baseline. Our analysis suggests that the feedback signal tends to modify neural selectivity towards co-linearity in both the end-zone and the side-zone.

**SDPC feedback signals modulate the activity within the interaction map.** To study the effect of the feedback on the level of activity within the interaction map, we introduce the ratio $r_a(k_{FB})$ between the activity with a certain feedback strength and the activity when the feedback is suppressed (see Eq 25 in section Model and methods). Coloring the interaction map using a color scale proportional to $r_a(k_{FB})$ allows us to identify which part of the map is more activated with the feedback. First, we observe qualitatively that the interaction map in the end-zone is more strongly activated when the feedback connection is active. On the contrary, the side-

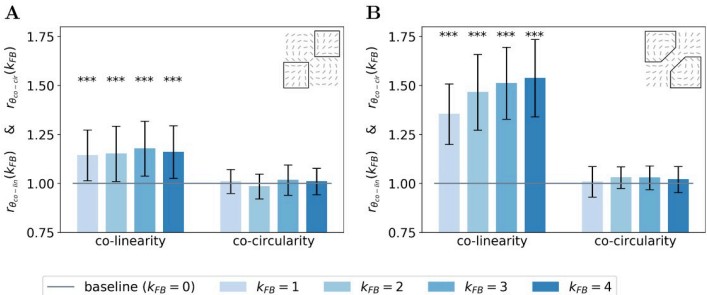

**Fig 6. Relative co-linearity and co-circularity of the V1 interaction map w.r.t. to feedback. (A)** In the end-zone. **(B)** In the side-zone. For each plot, the left and right block of bars represents the relative co-linearity (i.e. $r_{\theta_{co-lin}}(k_{FB})$) and co-circularity (i.e. $r_{\theta_{co-cir}}(k_{FB})$) with a feedback strength ranging from 1 to 4 w.r.t. their respective value without feedback (see Eqs 23 and 24). Bars' heights represent the median over all the orientations, and error bars are computed as the median absolute deviation. The baseline represents co-linearity / co-circularity when $k_{FB} = 0$.

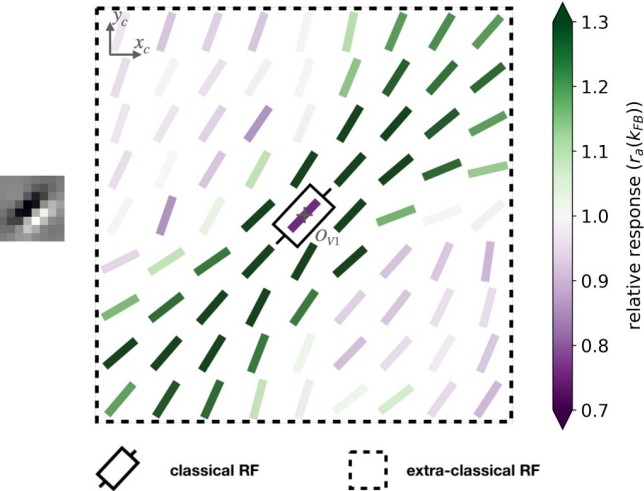

**Fig 7. Example of a 9 × 9 interaction map of a V1 area centered on neurons strongly responding to a central preferred orientation of 45˚, and colored with the relative response w.r.t. no feedback.** The feedback strength is set to 1 and the SDPC is trained on natural images. At each location identified by the coordinates $(x_c, y_c)$ the angle is $\bar{\theta}[x_c, y_c]$ (see Eq 11) and the color scale is proportional to $r_a(k_{FB})$ (see Eq 25). The color scale being saturated toward both maximum and minimum activity, all the activities above 1.3 or below 0.5 have the same color, respectively dark green or purple.

zone exhibited weaker activities when feedback is turned on (see Figs 7 and S4 for examples of this phenomenon with $k_{FB} = 1$). Note also that the activity in the center of the interaction map, which corresponds to the classical Receptive Field (RF) area, is lowered when feedback is active.

We now generalize, refine and quantify these qualitative observations. We include a third region of interest, the classical RF, to confirm the decreasing activity observed qualitatively at this location. We report the median of the ratio $r_a(k_{FB})$ over all central preferred orientations, for the end-zone, the side-zone, and the classical RF. This analysis is repeated for a feedback strength ranging from 1 to 4 (see Fig 8). We observe an increase of the activity in the end-zone of the interaction map with feedback compared to the end-zone of the interaction without feedback (see Fig 8A). This increase is significant as quantified by all pair-wise statistical Wilcoxon signed-rank tests with the baseline (WT($N = 51$, $p < 1e − 3$)). For larger feedback strengths, we observe a higher activity in the end-zone which is also significant (all pair-wise statistical tests between all feedback strengths (WT($N = 51$, $p < 0.01$)). For example, in the end-zone, the median activity over all the central preferred orientations is 16% and 25% higher with a respective feedback strength of 1 and 4 compared to the median when feedback is suppressed. This suggests that the feedback signals excite neurons in the end-zone of the interaction map. In contrast, we observe a slight decrease of activity in the side-zone of the interaction map with feedback active compared to when feedback is suppressed (see Fig 8B). The decrease compared to the baseline is significant for $k_{FB} = 1$ and $k_{FB} = 2$ (WT($N = 51$, $p < 1e − 2$)). For higher feedback strength, the lowered activity in the side zone becomes less significant. The activity in the classical RF exhibits a significant decrease compared to the baseline (WT($N = 51$, $p < 1e − 3$)). In addition, the larger the feedback strength, the weaker the activity in the center of the interaction map (WT($N = 51$, $p < 1e − 2$)). For example, we report a change in the decrease from −28% for $k_{FB} = 1$ to −34% for $k_{FB} = 4$ compared to the activity in the center of the interaction map without feedback (see Fig 8C).

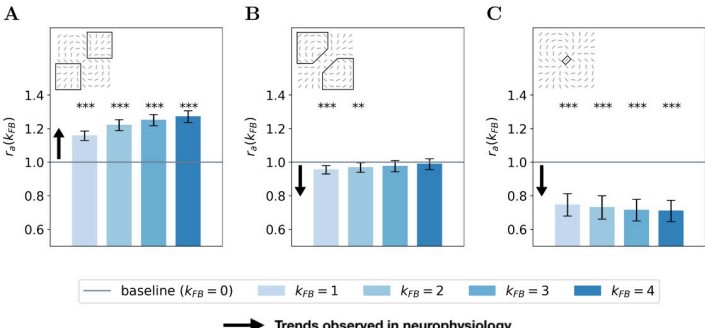

**Fig 8. Relative response of V1 interaction map w.r.t. no feedback for all central preferred orientations. (A)** In the end-zone. **(B)** In the side-zone. **(C)** In the center (i.e. classical RF). Bars' height represent the median over all the central preferred orientations, and error bars are computed as the median absolute deviation. The computation of the relative response, denoted $r_a(k_{FB})$, is detailed in Eq 25. The baseline represents the relative response without feedback. Black arrows represent the trends observed in neurophysiology (see section 'Comparing SDPC results with neurophysiology' for more details).

We report the spatial profile of the median activity along the axis of the central preferred orientation (see Fig 9). For all distances from the center, the activity along the central preferred orientation axis of interaction map is significantly higher than the activity without feedback (all pair-wise statistical tests with the baseline: $WT(N = 51, p < 1e − 2)$). The only exception is in the classical RF of the interaction map, where the activity is weaker when feedback is active (see also Fig 8C). This inhibition in the classical RF of the map compared to the baseline is significant as quantified with pair-wise statistical tests ($WT(N = 51, p < 1e − 3)$). Even if activities for $k_{FB} \neq 0$ along the central preferred orientation axis are always higher than the activity with $k_{FB} = 0$, they tend to decrease with distance to the center (pair-wise statistical test for different locations: $WT(N = 51, p < 1e − 2)$). Especially, for $k_{FB} = 4$, the neurons located just near the center exhibit a response +36% higher than the same neurons without feedback. With the

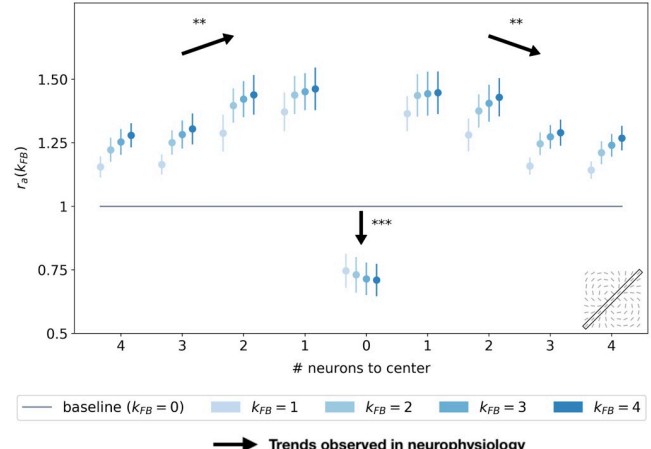

**Fig 9. Relative response w.r.t. no feedback along the axis of the central preferred orientation of V1 interaction map.** Each point represents the median over all the orientations, and error bars are computed as the median absolute deviation. The x-axis represents the distance, in number of neurons, to the center of the interaction map. The computation of the relative response, denoted $r_a(k_{FB})$, is detailed in Eq 25. The baseline represents the relative response without feedback. Black arrows represent the trends observed in neurophysiology (see section 'Comparing SDPC results with neurophysiology' for more details).

same feedback strength, this increase of activity w.r.t to no feedback is reduced to 15% when the neurons are located 4 neurons away from the center. At a given position different from the center, increasing the feedback strength significantly increases the activity as quantified by all pair-wise statistical test (WT($N = 51$, $p < 1e − 2$)).

Our results exhibit three different kinds of modulations in the interaction map due to feedback signals. First, the activity in the classical RF of the map is reduced with the feedback. Second, the activity in the end-zone, and more specifically along the axis of the central preferred orientation is increased with the feedback. Third, the activity in the side-zone is reduced with the feedback.

## Effect of the feedback at the representational level

After investigating the effect of feedback at the lowest level of neural organization, we now explore its functional and higher-level aspects. In particular, this subsection is dealing with the denoising ability of the feedback signal.

**Denoising abilities emerge from the feedback signals of the SDPC.**   To evaluate the denoising ability of the feedback connection, we feed the Sparse Deep Predictive Coding (SDPC) model with increasingly more noisy images extracted from the natural images and the face databases. Then, we compare the resulting representations ($\gamma_i^{\text{eff}}$) with the original (non-degraded) image. To do this comparison, we conduct two types of experiments: a qualitative experiment that visually displays what has been represented by the model (see Figs 10A and S5A), and a quantitative experiment measuring the similarity between representations of noisy and original images (see Fig 10B and 10C on natural images and Fig 11A and 11B on face database). These two experiments are repeated for a noise level ($\sigma$) ranging from 0 to 5 and a feedback strength ($k_{\text{FB}}$) varying from 0 to 4. The similarity between images is computed using the median structural similarity index [43] over 1200 and 400 images for the natural images and face database, respectively. The structural similarity index varies from 0 to 1 such that the more similar the images, the closer the index is to 1. For comparison, we include a baseline (see the black curves in Figs 10 and 11) which is computed as the structural similarity index between original and noisy images for different levels of noise. It is important to note that this experiment has been conducted without re-training the SDPC. Therefore, the network is trained on non-degraded natural images and has not been explicitly asked to denoise degraded images.

We first observe that whatever the feedback strength, the first layer representations of the original image (first row, column 2 to 6 in S5A Fig) are relatively similar to the input image itself. This observation is supported by a structural similarity index close to 0.9 for all feedback strengths ($\sigma = 0$ in Fig 10B for natural images and Fig 11A for faces). On the contrary, second layer representations look more sketchy and exhibit fewer details than the image they represent (first row, column 7 to 11 in S5A Fig). This is also quantitatively backed by a structural similarity index fluctuating around 0.4 for the natural images database ($\sigma = 0$ in Fig 10C) and 0.6 for the face database ($\sigma = 0$ in Fig 11B). Interestingly, when input images are corrupted with noise (i.e. when $\sigma \geq 1$), and whatever the feedback strength, first layer representations systematically exhibit higher structural similarity index than the baseline (Fig 10B for natural images and Fig 11A for faces). This denoising ability of the SDPC, even without feedback is significant as reported by the pair-wise Wilcoxon signed-rank tests with the baseline for both databases (WT($N = 1200$, $p < 1e − 2$) for natural images database, and WT($N = 400$, $p < 1e − 2$) for the face database). More importantly, the higher the feedback strength, the higher the similarity index. In particular, on the natural images database, when the input is highly degraded by noise ($\sigma = 5$), the similarity is 0.02 for the baseline, 0.03 for $k_{\text{FB}} = 0$, 0.05 for

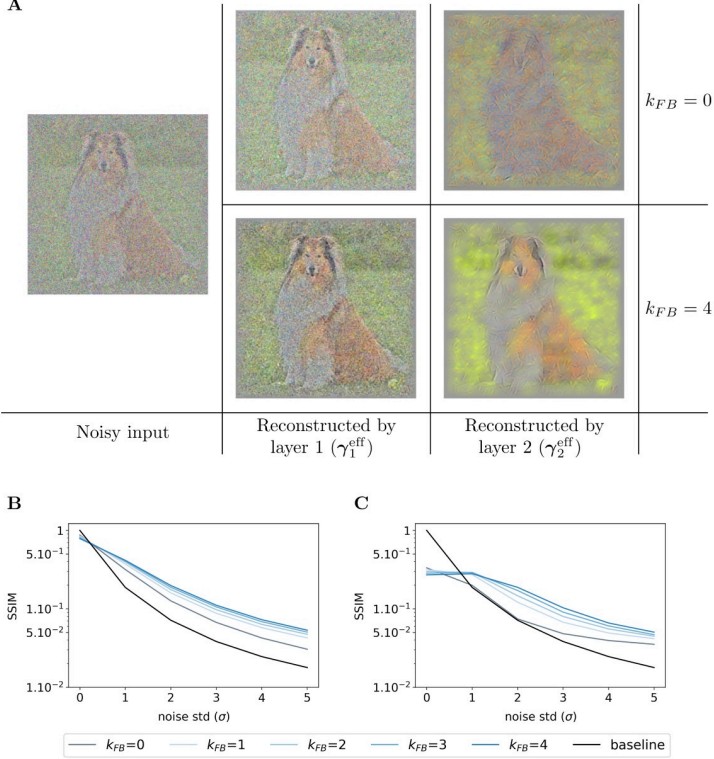

**Fig 10. Effect of the feedback strength on noisy images from natural images database. (A)** In the left column, one image is corrupted by Gaussian noise of mean 0 and a standard deviation of 2 ($\sigma$). The central column exhibits the representations made by the first layer ($\gamma_1^{\text{eff}}$), and the right-hand column the representations made by the second layer ($\gamma_2^{\text{eff}}$). Within each of these blocks, the feedback strength ($k_{\text{FB}}$) is equal to 0 in the top line and 4 in the bottom line. **(B)** We plot the structural similarity index (higher is better) between original images and their representation by the first layer of the SDPC. **(C)** We plot the structural similarity index between original images and their representation by the second layer of the SDPC. All curves represent the median structural similarity index over 1200 samples of the testing set and present a logarithmic scale on the y-axis. The color code corresponds to the feedback strength, from grey for $k_{\text{FB}} = 0$ to darker blue for higher feedback strength. The black line is the baseline, it is the structural similarity index between the noisy and original input images.

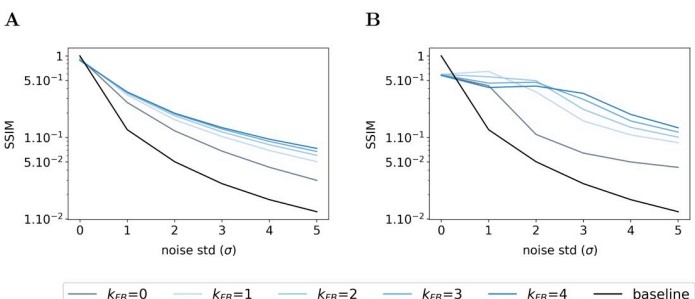

**Fig 11. Effect of the feedback strength on noisy images from face database.** This figure description is similar to the description of the Fig 10B and 10C. For the face database, all presented curves represent the median structural similarity index over 400 samples of the testing set.

$k_{FB} = 1$ and 0.06 for $k_{FB} = 4$ (see Fig 10B). This improvement of the denoising ability with higher feedback strength when $\sigma = 5$ is also significant as quantified by the pair-wise statistical tests between all feedback strengths (WT($N = 1200$, $p < 1e - 2$)). The inter-image variability of the structural similarity of first layer representation as quantified by the median absolute deviation is low compared to the median similarity on the natural images database (see S5D Fig). In the face database, for a highly degraded input ($\sigma = 5$), the structural similarity index is 0.01 for the baseline, 0.03 for $k_{FB} = 0$, 0.05 for $k_{FB} = 1$ and 0.07 for $k_{FB} = 4$. On the face database, the increases of the first layer similarity with the feedback strength when inputs are highly degraded ($\sigma = 5$) are significative as measured by all the pair-wise statistical tests between all feedback strengths (WT($N = 400$, $p < 1e - 2$)). Inter-image variability of the similarity of the first layer representation is also lower than the corresponding median on the face database (see S6C Fig). Our analysis suggests that the feedback connection in the Predictive Coding (PC) framework (i.e. when $k_{FB} > 0$) allows the first layer of the network to better denoise degraded images. It is interesting to mention again that this property is emergent as the network has never been explicitly trained to denoise degraded images.

**Effect of sparsity on denoising.** We have seen in the previous subsection that the feedback connection exhibited an emergent denoising ability. In this subsection, we wonder what is the effect of the sparsity on the denoising ability of the Sparse Deep Predictive Coding (SDPC). To answer this question, we feed the SDPC with increasingly blurred images, we vary the level of the sparsity in the first layer during inference (i.e. $\lambda_1$) and we assess its impact on the reconstruction quality using the structural similarity index. Figs 12 and 13 show the evolution of the similarity on the natural images and face databases and for 3 different levels of sparsity on the first layer: no sparsity at all (i.e. $\lambda_1 = 0$), intermediate sparsity (i.e. $\lambda_1 = 1.5$), and high sparsity (i.e. $\lambda_1 = 3.0$). We observe that high sparsity levels are beneficial for better reconstruction quality of the first layer when the input images are strongly degraded (see dark brown curves in Figs 12A and 13A). In contrast, when input images are not degraded at all (i.e. $\sigma = 0$), the lower the first layer sparsity the better the reconstruction quality (see light brown curves on Figs 12A and 13A). In addition, we observe a similar phenomenon for the second layer of the SDPC (see Figs 12B and 13B). Our analysis suggests that sparsity is playing a crucial role when it comes to denoise strongly degraded input images.

In this section, we conducted a qualitative and quantitative analysis of the denoising ability of Sparse Deep Predictive Coding (SDPC) model. Our results suggest that not only the feedback connection but also the sparse representation allows the SDPC to better recover images

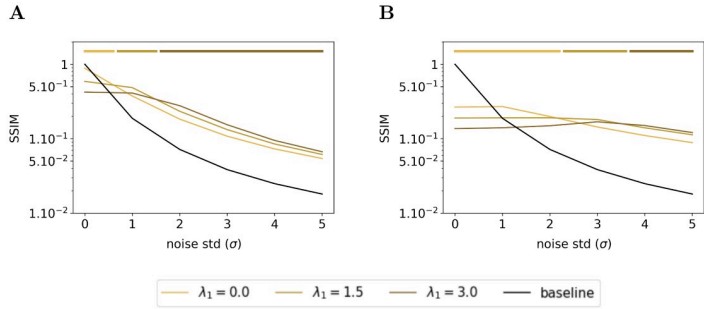

**Fig 12. Effect of the first layer sparsity on noisy images from natural images database with $k_{FB} = 4$. (A)** Structural similarity index between original images and their representation by the first layer of the SDPC. **(B)** Structural similarity index between original images and their representation by the second layer of the SDPC. Top lines represent the sparsity levels that maximize the similarity for different levels of degradation. All presented curves represent the median structural similarity index over 1200 samples of the testing set.

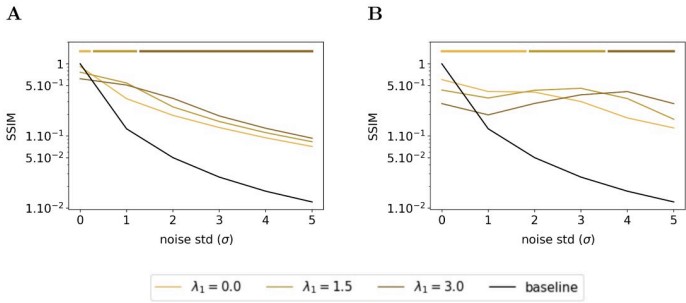

**Fig 13. Effect of the first layer sparsity on noisy images from face database with k$_{FB}$ = 4.** This figure description is similar to the description of the Fig 12. For the face database, all presented curves represent the median structural similarity index over 400 samples of the testing set.

degraded with noise. Therefore, the emergent denoising ability of the proposed model is directly deriving from the combination of the 2 components of the SDPC that are Sparse Coding and Predictive Coding. The superior denoising capacity of the $2^{nd}$ layer of the SDPC suggests that the network is able to disentangle informative features from noisy background. Such a disentangling mechanism might help the network to better recognize object when the input is corrupted.

## Discussion

Herein, we have conducted computational experiments on a 2-layered Sparse Deep Predictive Coding (SDPC) model. The SDPC leverages feedforward and feedback connections into a model combining Sparse Coding (SC) and Predictive Coding (PC). As such, the SDPC learns the causes (i.e. the features) and infers the hidden states (i.e. the activity maps) that best describe the hierarchical generative model giving rise to the visual stimulus (see Fig 14 for an illustration of this hierarchical model and Eq 1 for its mathematical description).

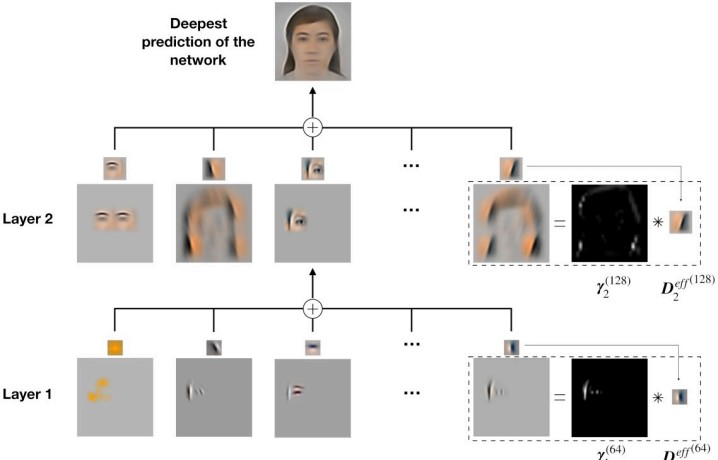

**Fig 14. Illustration of the hierarchical generative model learned by the SDPC model on the face database.** The deepest prediction (first row) is viewed as the sum of the features prediction (the second row). These feature predictions are computed as the convolution between one channel of $\gamma_2$ and the corresponding features in $D_2$. Similarly, the eyes can be decomposed using $\gamma_1$ and $D_1$(the third row).

We use this model of the early visual cortex to assess the effect of the early feedback connection (i.e feedback from V2 to V1) through different levels of analysis. At the neural level, we have shown that feedback connections tend to recruit more neurons in the first layer of the SDPC. We have introduced the concept of interaction maps to describe the neural organization in our V1-model. Interestingly, the interaction maps generated when natural images are presented to the model are very similar to biologically observed association fields. In addition, interaction maps allow us to describe the neural reorganization due to feedback signals. In particular, we have observed that feedback signals align neurons co-linearly to the central preferred orientation. At the activity level, we observe three different kinds of feedback modulatory effects. First, the activity in the classical Receptive Field (RF) is decreased. Second, the activity in the end-zone of the extra-classical RF and more specifically along the axis of the central preferred orientation is increased. Third, the activity in the side-zone of the extra-classical RF is reduced. At the representational level, we have investigated the role of feedback signals when input images are degraded using Gaussian noise. We have demonstrated that higher feedback strengths allow better denoising ability. We have also shown that sparsity plays a crucial role to recover degraded images. In this section, we link our model with the original PC model [29] and we interpret our computational findings in light of current neuroscientific knowledge.

## SDPC extends Rao & Ballard's PC model

The Sparse Deep Predictive Coding (SDPC) model is directly inspired by the Predictive Coding (PC) model proposed by Rao & Ballard [29] and extends it to a scale that is more realistic for cortical processing in the visual cortex. The original PC model had few dozens of neurons (20 in the first layer and 32 in the second one), linked with fully-connected synapses and trained on patches extracted from 5 natural images. In this work, the SDPC leverages hundreds of thousands of neurons ($\approx 5 \times 10^5$ neurons in the first layer, $\approx 8 \times 10^5$ neurons in the second layer for the network trained on natural images) and convolutional synaptic weights trained on thousands of natural images. In terms of analysis, the interaction maps we introduced confirm and extend the results from Rao & Ballard. Our model also described the end-stopping effects inside the classical Receptive Field (RF): we observed a strongly decreased activity in the classical RF for extended contours when the feedback connection was activated (see Fig 5 from [29] and see Figs 8C and 9). Last but not least, the convolutional framework of the SDPC allows us to extend the Rao & Ballard findings beyond the classical RFs and to observe that feedback signals play a role in the extra-classical RF. It tends to reinforce neural activity along the preferred orientation axis (see Figs 9 and 7) and to reshape neural selectivities to better reflect association fields.

## SDPC learns cortex-like RFs while performing neuro-plausible computation

The Sparse Deep Predictive Coding (SDPC) model satisfies some of the computational constraints that are thought to occur in the brain, notably local computations [44]. The locality of the computation is ensured by Eq 5: the new state of a neural population (whose activity is represented by $\gamma_i^{t+1}$) only depends on its previous state ($\gamma_i^t$), the state of adjacent layers ($\gamma_{i-1}^t$ and $\gamma_{i+1}^t$) and the associated synaptic weights ($D_i$ and $D_{i+1}$). In the SDPC we have used the convolutional framework to enforce a retinotopic organization of the activity map. The convolutional operator suggests that features are shared at every position of the activity map. This assumption has the advantage to model the position invariance of RFs observed in the brain. Nevertheless, the weight-sharing mechanism is far from being bio-plausible. Interestingly, recent

studies have shown that imposing local RFs to fully connected synapses allows to mimic convolutional features without enforcing the weight-sharing mechanism [45]. Therefore, it suggests that convolution-like operations might be implemented in the brain in the form of locally-connected synapses. All these neuro-plausible constraints we have included in the SDPC makes it unique compared to frameworks like feedforward neural networks or auto-encoders. These networks are trained using a global loss function minimized through back-propagation and do not leverage top-down signals during the inference process. Not only the processing but also the result of the training exhibits tight connections with neuroscience. The first-layer Receptive Fields (RFs) (Fig 2B for the natural images database and Fig 2F for the face database) are similar to the V1 simple-cells RFs, which are oriented Gabor-like filters [46, 47]. Olshausen & Field have already demonstrated, in a shallow network, that oriented Gabor-like filters emerge from sparse coding strategies [25], but to the best of our knowledge, this is the first time that such filters are exhibited in a 2-layers network combining neuro-plausible computations with Predictive Coding and Sparse Coding (in [29], Rao & Ballard exhibited only first layer filters on their model constrained with a sparse prior). This architecture allows us to observe an increase in the specificity of the neuron's RFs with the depth of the network. This observation is even more striking when the SDPC is trained on the face database, which presents less variability compared to the natural images database. On face images, second layer RFs exhibit features that are highly specific to faces (eyes, mouth, eyebrows, contours of the face). Interestingly, it was demonstrated with neurophysiological experiments that neurons located in deeper regions of the central visual stream are also sensitive to that particular face features [48, 49].

## Comparing SDPC results with neurophysiology

At the electrophysiological level, it has been demonstrated that as early as in the V1 area, feed-back connections from V2 or V4 could either facilitate [12] or suppress [16] lateral interactions. These modulations help V1 neurons to integrate contextual information from a larger part of the visual field and play a causal role in increasing and decreasing activity for neurons encoding for the contour and the background, respectively [18, 50, 51]. The SDPC model behaves similarly: i) Fig 8A is showing a feedback-dependent increase of activity for neurons located in the end-zone (i.e. in the direction of the contour) and ii) Fig 8B exhibits a feedback-dependent decrease activity of neurons located in the side-zone (i.e. in the direction of the background). As mentioned in the previous subsection, the SDPC is also consistent with the increase of end-stopping effect related to the increase of the feedback strength. In electrophysi-ology these phenomenons has been observed in monkeys using attentional modulations (that we will interpret as a modulation of the feedback strength) [52] or by cooling-down areas located after V1 to remove the feedback signals [17]. Moreover, it has been demonstrated that the neural excitation due to feedback signal from V4 to V1 on neurons located on contours was strongly dependent on the length of the contours [18]. An extended contour triggered smaller extra-feedback signals compared to shorter contours (see Fig 2A in [18]). This electrophysiological observation is in line with the SDPC results shown in Fig 9: neurons located along the axis of the contour but far away from the classical Receptive Field (RFs) are less strongly excited than those closer to the classical RF.

## Functional interpretation of the observed V1 interaction maps

It was assumed that association fields were represented in V1 to perform such a contour inte-gration [21]. Interestingly, SDPC first-layer interaction maps exhibit a co-linear and co-circu-lar neural organization very similar to association fields even without feedback (see Fig 5). We

formulate the hypothesis that this specific organization is mainly related to the statistics of edge co-occurrences in natural images [20]. Nevertheless, the modulation of neural activity within the interaction map mediated by feedback goes towards a better contour integration. Indeed, the increase of activity in the end-zone and the decrease of activity in the side-zone seem to be optimal to integrate smooth and close contours [53] (see Figs 7–9). In addition, the organizational feedback modulation in the interaction map reveals that feedback signals tend to reorganize the side-zone to promote orientations that are co-linear to the central preferred orientation (see Fig 6). This organization may provide an optimal substrate to integrate dynamic stimuli along two axes: a parallel one with the apparent motion-like sequence of oriented stimuli moving along the end-zone direction [41, 54, 55], and a perpendicular one for oriented stimuli moving perpendicular to their orientation. Interestingly, oriented Gabors moving in an apparent sequence along the parallel axis are perceived as faster as Gabors moving along the orthogonal axis [55]. Furthermore, aligning the side-zone region to the preferred orientation could also contribute to the aperture problem [56] and the observed bias for perceiving oriented bars as moving in the direction perpendicular to their orientation [57, 58].

## Do lateral interactions increase the sparseness of neural activity in V1?

In this paper, we have assumed that recurrent internal processing could be modeled using sparse coding. Is it a realistic hypothesis? One of the main roles of sparse coding is to enforce competition among neurons: it suppresses weakly activated neurons to promote strongly activated ones. In other words, sparse coding performs "explaining away". Interestingly, when 2 stimuli (blobs) were presented at different locations and timings, it has been observed in monkeys' area V1 that a suppressive wave tends to spatially disambiguate the positions of the 2 stimuli [59]. This effect was attributed to lateral interactions (due to the spatio-temporal properties of the effect) and can be thought of as an explaining away mechanism. Other studies have also demonstrated that lateral interactions exacerbate competition in cortical columns with different orientations or ocular dominance [60–62]. Therefore, our sparse coding model accounts for one possible function of lateral interaction. Nevertheless, the sparse coding algorithm we use (i.e. the Fast Iterative Soft-Thresholding Algorithm (FISTA), see section "Model and methods") doesn't allow to explicitly learn a lateral connectivity matrix. Consequently, one might consider other sparse coding algorithms including lateral connection weights to provide a more accurate model of cortical columns [63].

## SDPC accounts for object processing in V1 with degraded images

In this paper, we demonstrate that both the feedback connection introduced by Predictive Coding (PC) as well as Sparse Coding (SC) allow the Sparse Deep Predictive Coding (SDPC) to denoise the representation generated in the first and the second layer. Interestingly, this is an emergent property of the network as we do not explicitly train the SDPC to denoise the input. This crucial point makes the SDPC very different from denoising sparse auto-encoder that are trained to reconstruct corrupted input. Psychophysical experiments using backward masking demonstrated that categorization performances were substantially impaired when a mask followed a highly degraded stimulus (by occlusion or contrast reduction) [24, 64]. This suggests that feedback is crucial to recognize a degraded image. We demonstrate a similar result by assessing the representations of the first layer of the SDPC when the model is fed with increasingly more noisy images and with different feedback strengths. In particular, we have shown that feedback connections from V2 to V1 have the ability to denoise corrupted images (see Figs 11, 12, S5 and S6). In addition, the previously mentioned psychophysical studies suggest that feedback connections are not bringing any change in recognition accuracy when

non-degraded images are presented to the subjects [24, 64]. In contrast, the structural similarity index between the original image and the first layer representation when the SDPC is fed with a non-degraded image exhibits a slight decrease, but significant enough, when the feedback strength or the sparsity is increased. We formulate the hypothesis that this discrepancy is mainly coming from the fixed value we give to the feedback strength and to the sparsity parameter. In the brain, representations are strongly subject to attentional modulation, and a recent study has suggested that attention can be understood as a mechanism weighting feedback connections using the level of uncertainty [65]. Therefore, one might consider replacing the parameters $k_{FB}$ and $\lambda_i$ by internal state variables that would be specific to each input image (similar to the sparse maps $\gamma_i$). For example, if the input image is not degraded, the feedback signals should be weak as no higher-layer information is needed to faithfully represent the sensory input. On the contrary, if the input is strongly degraded, the feedback connection should be strong enough to bring additional information from higher-layer to compensate for the high uncertainty in the first layer representations. Such modifications should allow the SDPC to adapt to the specificity of each input, and could be used to model attentional mechanisms in the SDPC.

## Concluding remarks

In this study, we have shown that the first layer of the Sparse Deep Predictive Coding (SDPC) model represents the visual input similarly to V1. We have also demonstrated that feedback from V2 may modulate the interaction map in such a way to promote contour integration. This improvement in contour integration with feedback strength resulted in a better representation when noisy images were presented to the SDPC. Note that the proposed SDPC is a simplified version of perceptual inference models based on free-energy optimization [66, 67]. While free-energy estimates the entire distribution of error and prediction signals, our SDPC only assesses their most likely values. One interesting perspective would be to extend the SDPC to make it fit the precision-weighted message passing implemented in the free-energy framework. Another interesting perspective would involve building deeper SDPC networks to model brain areas like V4 or IT. In such a case, we expect that feedback signals in a deeper layer should highlight higher-level concepts (e.g. the global shape of an object or its identity-related items). In general, we foresee great perspectives to such a description of the brain both in computational neuroscience to understand perceptual mechanisms and in artificial intelligence for tasks like denoising, classification or inpainting.

## Model and methods

In this section, we detail the Sparse Deep Predictive Coding (SDPC) model. We first explain how the SDPC is directly related to the Predictive Coding (PC) theory. Next, we describe the mathematics behind the inference and the learning process. We then explicitly describe the back-projection mechanism used to interpret and visualize inference and learning results. We also describe the databases and the network parameters we adopted to train the SDPC. Finally, we detailed all the calculations needed to generate interaction maps.

### SDPC model

**Variable dimensionality and convolutional operator.** In the entire article, we have adopted the machine learning convention to describe the dimensions of our dictionary $D$ and of the activity map $\gamma$: a dictionary $D$ of size $[n_f, n_c, w, h]$ coud be interpreted as a collection of $n_f$ features of size $n_c \times w \times h$. $n_c$ is then the number of channels of the representation on which we apply the convolution. For example, in the case of the first layer dictionary, $n_c = 1$ for

grayscale images and $n_c = 3$ for color images (i.e. the natural images database). The width and height are denoted by $w$ and $h$, respectively. Similarly, an activity map $\gamma$ of size $[n_f, w_m, h_m]$ could be interpreted as a collection of $n_f$ 2D maps of dimension $(w_m, h_m)$.

Using the same dimension notation, we define our discrete 2D-convolutional operator using the following operation (see Eq 13)

$$\gamma_{i-1} \qquad = \boldsymbol{D}_i^T * \gamma_i$$

$$\text{with} \quad \gamma_{i-1}[j, k, l] \quad = \sum_{m=1}^{n_c}\sum_{p=1}^{w}\sum_{q=1}^{h} \boldsymbol{D}_i^T[j, m, p, q] \times \gamma_{i-1}[m, k-p, l-q] \tag{13}$$

$$s.t. \ k - p \in [\![1, w_m]\!] \ \text{and} \ l - q \in [\![1, h_m]\!]$$

For the sake of clarity, in the mathematical description of our model, we replaced the convolution by a matrix-vector product. This mathematical transformation is valid as one can always find an operator that transforms the dictionary $\boldsymbol{D}_i$ into a Toeplitz matrix (we denote this operator $\mathcal{T}$ in the following equation):

$$\gamma_{i-1} = \mathcal{T}(\boldsymbol{D}_i^T) \times \gamma_i = \boldsymbol{D}_i^T * \gamma_i \ \text{s.t.} \ \mathcal{T} \ \text{is the transformation of} \ \boldsymbol{D}_i^T \ \text{into a Toeplitz matrix} \tag{14}$$

In Eq 14, the sign $\times$ denotes the matrix-vector multiplication. To facilitate the reading of the mathematical equation, we have purposely abuse the notation in the paper such that: $\gamma_{i-1} = \boldsymbol{D}_i^T\gamma_i = \mathcal{T}(\boldsymbol{D}_i^T) \times \gamma_i = \boldsymbol{D}_i^T * \gamma_i$.

**From predictive coding to sparse deep predictive coding.** Fig 1 shows the architecture of a 2-layered SDPC model that takes an image $\boldsymbol{x}$ as an input. As the SDPC is relying on the Predictive Coding (PC) theory [29], it is continuously generating top-down predictions such that the neural population at one level ($\gamma_i$) predicts the neural activity at the lower level ($\gamma_{i-1}$). The prediction from a higher level is sent through a feedback connection to be compared to the actual neural activity. This elicits a prediction error, $\boldsymbol{\epsilon}_i$, that is forwarded to the following layer to update the population activity towards improved prediction. This dynamical process repeats throughout the hierarchy until the bottom-up process no longer conveys any new information. We force the weights of the feedforward connection ($\mathbf{D}_i$) to be reciprocal to the weights of the feedback connection ($\mathbf{D}_i^T$) [29, 68]. We also impose a convolutional structure to $\mathbf{D}_i$ to strengthen the proximity with the overlapping Receptive Fields (RFs) observed in the visual cortex. Mathematically, the SDPC solves the hierarchical inverse problem formulated in Eq 1 by minimizing the loss function $\mathcal{L}$ defined in Eq 15. This optimization process is separated into two different but related steps: inference and dictionary learning. The inference process involves finding a sparse activity map of the input considering the synaptic weights are fixed. Once the activity map has been estimated, the next step is to update the synaptic weights to better fit the dataset. We iterate these two processes until the convergence is reached.

**Inference.** To obtain a convex cost, we relax the $\ell_0$ constraint in Eq 1 into a $\ell_1$-penalty. It defines, therefore, a loss function that could be minimized using first-order methods like Iterative Shrinkage Thresholding Algorithms (ISTA) [69]. This algorithm is proven to be computationally cheap and offers fast convergence rate. In practice, we use an accelerated version of this algorithm called the Fast Iterative Soft Thresholding Algorithm (FISTA). Eq 15 describes the generalized loss function, that is minimized at each layer using the Iterative Soft Thresholding Algorithm.

$$\mathcal{L}_i = \frac{1}{2} \parallel \gamma_{i-1} - \mathbf{D}_i^T\gamma_i \parallel_2^2 + \frac{k_{\text{FB}}}{2} \parallel \gamma_i - \mathbf{D}_{i+1}^T\gamma_{i+1} \parallel_2^2 + \lambda_i \parallel \gamma_i \parallel_1 \tag{15}$$

One inference step used to update $\gamma_i$ is shown in Eq 5. In Eq 5, $\mathcal{T}_\alpha$ denotes a non-negative soft thresholding operator, as defined in Eq 16. $\eta_{c_i}$ is the learning rate of the inference process, it is computed as the inverse of the largest eigenvalue of $\mathbf{D}_i^T \mathbf{D}_i$ [69].

$$\mathcal{T}_\alpha(x) = \begin{cases} x - \alpha & \text{if } x \geq \alpha \\ 0 & \text{if } x \leq \alpha \end{cases} \tag{16}$$

Fig 1 shows how we can interpret the update scheme described in Eq 5 as one loop of the inference process of a recurrent layer. This recurrent layer forms the building block of the Sparse Deep Predictive Coding (SDPC) network (see S1 Algo for the complete pseudo-code of the SDPC inference process). We initialize all the activity maps $\gamma_i^t$ to zero at the beginning of the inference process. We consider the inference process is finalized once all the activity maps have reached a fixed point. Our fixed point consists in a threshold ($T_{fp}$) on the relative variation of each activity map (Eq 17).

$$\gamma_i^t \text{ has reached a fixed point if } \frac{\| \gamma_i^t - \gamma_i^{t-1} \|_2}{\| \gamma_i^t \|_2} < T_{fp} \tag{17}$$

**Dictionary learning.** The SDPC learns the synaptic weights using a stochastic gradient descent on $\mathcal{L}_i$. Eq 6 describes one step of the dictionary learning process.

In Eq 6, $\mathbf{D}_i^t$ is the set of synaptic weights at time step $t$ and $\eta_{L_i}$ is its learning rate. At the beginning of the learning, all weights are initialized using the standard normal distribution (mean 0 and variance 1). The learning step takes place after the convergence of the inference process is achieved (see Algo 1). It was demonstrated that this alternation of inference and learning offers a reasonable convergence guarantee [37]. After every dictionary learning step we $\ell_2$-normalize each weight to avoid any redundant solution.

**Algorithm 1:** Alternation of inference and learning

```
while convergence not reached do
    for i = 1 to L do
        γᵢᵗ⁺¹ = 𝒯_{η_{cᵢ}λᵢ}(γᵢᵗ − η_{cᵢ}∇_{γᵢᵗ}ℒᵢ)        # inference
    for i = 1 L to L do
        Dᵢᵗ⁺¹ = Dᵢᵗ − ηᵢ∇_{Dᵢᵗ}ℒᵢ                        # learning
```

**Back-projection mechanism.** Interestingly, the dictionaries could be used to project (or back-project) the activity of a neural population and their associated synaptic weights into the next (or previous) level. Due to their high dimensionality, the weights $\mathbf{D}_i$ are difficult to interpret and visualize for $i > 1$ as they represent a structure into an intermediate feature space at layer $i - 1$. To overcome this limitation, we back-project the weights $\mathbf{D}_i$ into the input space, which is the visual space [37]. This back-projection, called effective dictionary and denoted by $\mathbf{D}_i^{\text{eff}}$, could be interpreted as the set of Receptive Fields (RFs) of the neurons located in layer i. Note that the network doesn't directly compute the effective dictionaries, and it is used only to visualize what has been learned and represented by the model. Mathematically, the effective dictionaries are described in Eq 3, and illustrated in S7 Fig. Similarly, we defined $\gamma_i^{\text{eff}}$ as the back-projection into the visual space of the hidden states variable $\gamma_i$ (Eq 18). This mechanism is used to reconstruct the input image from one intermediate layer.

$$\gamma_i^{\text{eff}} = \mathbf{D}_i^{\text{eff}^T} \gamma_i \tag{18}$$

## Databases

We train our SDPC model on two different databases: The Chicago Face Database (CFD) [70] and STL-10 [71].

**The Chicago Face Database** consists of 1, 804 high-resolution (2, 444 × 1, 718 px), color, standardized photographs of male and female faces with varying ethnicity between the ages of 18 and 40 years. We re-sized the pictures to 170 × 120 px to keep reasonable computational time. This database is partitioned into batches of 10 images. This dataset is split into a training set composed of 721 images and a testing set of 400 images. No validation set was used.

**The STL-10 database** is a recognition dataset developed for unsupervised feature learning and composed of color photographs with a resolution of 96 × 96 px representing animals (bird, cat, deer, dog, horse, monkey) and non-animals (airplane, car, ship, truck). The images are highly diverse (different viewpoints, backgrounds, . . .) and could be considered as natural images. The set is partitioned into a training set of 5000 images and a testing test of 1200 images. No validation set was used.

All the curves, images and histograms presented in this paper are generated using the testing set. The training set is used only to learn the synaptic weights. All these databases are pre-processed using local contrast normalization and whitening. Local contract normalization is inspired by neuroscience and consists in a local subtractive and divisive normalization [40]. In addition, we use whitening to reduce dependency between pixels.

## Network parameters

Networks and training parameters of the Sparse Deep Predictive Coding (SDPC) model are summarized in Table 1 for the natural image and face databases. We used PyTorch 1.0 [72] to implement, train, and test the SDPC model.

## Interaction maps analysis

**Computation of the relative co-linearity and co-circularity for different feedback strength.** We measure the co-linearity deviation of the interaction map with a circular difference between the central preferred orientation ($\theta_c$) and the orientation of the interaction map (see Eq 19). The co-circularity deviation is quantified using a circular difference between a map of orientations that are co-circular to the central preferred orientation and the angle of

**Table 1. SDPC network and training parameters on natural image and face databases.** The size of the convolutional kernels for each layer are shown in the format: [number of features, number of channels, width, height] (value of the convolutional stride).

|  |  | DataBase | |
|---|---|---|---|
|  |  | **Face images** | **Natural images** |
| network param. | $D_1$ size | [64, 3, 9, 9] (3) | [64, 3, 9, 9] (2) |
|  | $D_2$ size | [128, 64, 9, 9] (1) | [128, 64, 9, 9] (1) |
|  | $\lambda_1$ | 0.3 | 0.4 |
|  | $\lambda_2$ | 1.6 | 1.2 |
|  | $T_{stab}$ | 5e-3 | 5e-3 |
| training param. | # epochs | 250 | 250 |
|  | $\eta_{L_1}$ | 1e-4 | 1e-4 |
|  | $\eta_{L_2}$ | 5e-3 | 5e-3 |
|  | momentum | 0.9 | 0.9 |

the interaction map (see Eq 20) [73]. We simplify the calculation of the co-circular angle map in Eq 20 by centering the coordinate $(x_c, y_c)$ in the middle of the interaction map (the co-circular map is shown in the top right corner of the Fig 6A and 6B).

$$\boldsymbol{\theta}_{co-lin}[x_c, y_c] \quad = |\theta_c - \bar{\boldsymbol{\theta}}[x_c, y_c]| \tag{19}$$

$$\boldsymbol{\theta}_{co-cir}[x_c, y_c] \quad = |\operatorname{atan}\left(\frac{y_c - y_{co}}{x_c - x_{co}}\right) + \frac{\pi}{2} - \bar{\boldsymbol{\theta}}[x_c, y_c])|$$

$$\text{with } x_{co} \quad = \frac{\sin(\theta_c) \cdot (x_c^2 + y_c^2)}{2(\sin(\theta_c) \cdot x_c - \cos(\theta_c) \cdot y_c)} \tag{20}$$

$$\text{and } y_{co} \quad = \tan\left(\theta_c + \frac{\pi}{2}\right) \cdot x_{co}$$

The *atan* operator in Eq 20 denotes the arctangent operator. For a given feedback strength $k_{\text{FB}}$, we introduce two ratios, denoted $\tilde{\boldsymbol{\theta}}_{co-lin}^{k_{\text{FB}}}$ and $\tilde{\boldsymbol{\theta}}_{co-cir}^{k_{\text{FB}}}$ (see Eqs 21 and 22, respectively) to scale the co-linearity and co-circularity w.r.t. their marginal measure. In those equations, the marginal co-linearity ($\boldsymbol{\theta}_{co-lin}^{k_{\text{FB}}}[x_{\sim c}, y_{\sim c}]$) and co-circularity ($\boldsymbol{\theta}_{co-cir}^{k_{\text{FB}}}[x_{\sim c}, y_{\sim c}]$) correspond to the co-linearity and co-circularity computed outside of the contour neighborhood. To facilitate the interpretation of these ratios, we make sure they are following the same evolution than a precision measure. For example, if an interaction map exhibits a higher co-linearity with the central preferred orientation, then the corresponding $\tilde{\boldsymbol{\theta}}_{co-lin}^{k_{\text{FB}}}$ will be necessarily over 1.

$$\tilde{\boldsymbol{\theta}}_{co-lin}^{k_{\text{FB}}} = \frac{\boldsymbol{\theta}_{co-lin}^{k_{\text{FB}}}[x_{\sim c}, y_{\sim c}]}{\boldsymbol{\theta}_{co-lin}^{k_{\text{FB}}}[x_c, y_c]} \tag{21}$$

$$\tilde{\boldsymbol{\theta}}_{co-cir}^{k_{\text{FB}}} = \frac{\boldsymbol{\theta}_{co-cir}^{k_{\text{FB}}}[x_{\sim c}, y_{\sim c}]}{\boldsymbol{\theta}_{co-cir}^{k_{\text{FB}}}[x_c, y_c]} \tag{22}$$

To quantify how more co-linear and co-circular were the different regions of the interaction maps with feedback compared to the same interaction maps without feedback we used the following ratios (see Eqs 23 and 24):

$$\boldsymbol{r}_{\theta_{co-lin}}(k_{\text{FB}}) = \frac{\tilde{\boldsymbol{\theta}}_{co-lin}^{k_{\text{FB}}}}{\tilde{\boldsymbol{\theta}}_{co-lin}^{k_{\text{FB}}=0}} \tag{23}$$

$$\boldsymbol{r}_{\theta_{co-cir}}(k_{\text{FB}}) = \frac{\tilde{\boldsymbol{\theta}}_{co-cir}^{k_{\text{FB}}}}{\tilde{\boldsymbol{\theta}}_{co-cir}^{k_{\text{FB}}=0}} \tag{24}$$

**Computation of the relative activity with or without feedback.** To compare the relative activity with or without feedback, we introduce the ratio $r_a(k_{\text{FB}})$ (see Eq 25).

$$r_a(k_{\text{FB}}) \quad = \frac{|\bar{\boldsymbol{a}}(k_{\text{FB}})|}{|\bar{\boldsymbol{a}}(k_{\text{FB}} = 0)|} \tag{25}$$

## Supporting information

**S1 Fig. Receptive Fields (RFs) when the SDPC is trained on the natural images database.**
**(A)** 64 first layer RFs, sorted by activation probability in a descending order. The size of the

RFs is $9 \times 9$ px. **(B)** 128 second layer RFs, sorted by activation probability in a descending order. The size of the RFs is $22 \times 22$ px. All the visualized RFs are generated using Eq 3.
(TIF)

**S2 Fig. Receptive Fields (RFs) when the SDPC is trained on the face database. (A)** 64 first layer RFs, sorted by activation probability in a descending order. The size of the RFs is $9 \times 9$ px. **(B)** 128 second layer RFs, sorted by activation probability in a descending order. The size of the RFs is $33 \times 33$ px. All the visualized RFs are generated using Eq 3.
(TIF)

**S3 Fig. Example of $9 \times 9$ association field of V1 centered on neurons strongly responding to 6 different contour orientations, when the SDPC is trained on the natural image database.** From left to right and top to bottom the contour orientations are 0˚ **(A)**, −30˚ **(B)**, −60˚ **(C)**, 90˚ **(D)**, 60˚ **(E)** and 30˚ **(F)**. The feedback strength is set to 1. At each location identified by the coordinates $(x_c, y_c)$ the angle is $\bar{\theta}[x_c, y_c]$ (see Eq 11) and the color scale is $|\bar{a}[x_c, y_c]|$ (see Eq 12). The color scale being saturated toward both maximum and minimum activity, all the activities above 0.8 or below 0.3 have the same dark green or white color, respectively.
(TIF)

**S4 Fig. Example of a $9 \times 9$ association field in V1 colored with relative response w.r.t no feedback, centered on neurons strongly responding to 6 different contour orientations, when the SDPC is trained on the natural image database.** The feedback strength is set to 1. From left to right and top to bottom the contour orientations are 0˚ **(A)**, −30˚ **(B)**, −60˚ **(C)**, 90˚ **(D)**, 60˚ **(E)** and 30˚ **(F)**. At each location identified by the coordinates $(x_c, y_c)$ the angle is $\bar{\theta}[x_c, y_c]$ (see Eq 11) and the color scale is proportional to $r_a(k_{FB})$ (see Eq 25). The color scale being saturated toward both maximum and minimum activity, all the activities above 1.3 or below 0.5 have the same dark green or dark purple color, respectively.
(TIF)

**S5 Fig. Effect of the feedback strength on noisy images from the natural images database. (A)** In the left block, one image is corrupted by Gaussian noise of mean 0 and standard deviation ($\sigma$) varying from 0 to 5. The central block exhibits the representations made by the first layer ($\gamma_1^{\text{eff}}$), and the right-hand block the representations made by the second layer ($\gamma_2^{\text{eff}}$). Within each of these blocks, the feedback strength ($k_{FB}$) is ranging from 0 to 4 in columns. Highlighted images with black square are those selected in Fig 10. **(B)** median structural similarity index between 1200 original images and their reconstructions by the first layer of the SDPC. **(C)** Structural similarity index between original images and their reconstructions by the second layer of the SDPC. **(D)** Error, as computed with the median absolute deviation, of the structural similarity index plotted in **(B)** (i.e. for the first layer). **(E)** Error, as computed with the median absolute deviation, of the similarity index plotted in **(C)** (i.e. for the second layer). The color code corresponds to the feedback strength, from light grey for $k_{FB} = 0$ to darker blue for higher feedback strength. The black line is the baseline, it is the similarity between noisy and original input image.
(TIF)

**S6 Fig. Effect of the feedback strength on noisy images from the face database.** This figure description is similar to the description of the S5 Fig. For this database, all presented curves represent the median structural similarity index over 400 samples of the testing set.
(TIF)

**S7 Fig. Illustration of the back-projection mechanism.** The projection of the second layer dictionary into the visual space $(\mathbf{D}_2^{\text{eff}^T})$ is obtained by convolving the transpose of the first layer dictionary $(\mathbf{D}_1^T)$ by the second layer dictionary $(\mathbf{D}_2^T)$ [37]. This mechanism could be also used to back-project any activity map into the visual space (see Eq 18).
(TIF)

**S1 Algo. SDPC inference algorithm.** Pseudo-code of the inference using python-like pseudo algorithm. $\mathcal{T}_\alpha(\cdot)$ denotes the element-wise non-negative soft-thresholding operator. A fortiori, $\mathcal{T}_0(\cdot)$ is a rectified linear unit operator. # comments are comments.
(TIF)

## Author Contributions

**Conceptualization:** Victor Boutin.

**Formal analysis:** Victor Boutin, Frederic Chavane, Laurent Perrinet.

**Funding acquisition:** Franck Ruffier, Laurent Perrinet.

**Investigation:** Victor Boutin.

**Methodology:** Victor Boutin.

**Resources:** Laurent Perrinet.

**Software:** Angelo Franciosini.

**Supervision:** Frederic Chavane, Franck Ruffier, Laurent Perrinet.

**Validation:** Frederic Chavane, Franck Ruffier, Laurent Perrinet.

**Visualization:** Victor Boutin.

**Writing – original draft:** Victor Boutin.

**Writing – review & editing:** Victor Boutin, Angelo Franciosini, Frederic Chavane, Franck Ruffier, Laurent Perrinet.

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
