## [Decision Letter · Decision Letter 0]

25 Feb 2020

Dear Mr. Boutin,

Thank you very much for submitting your manuscript "Sparse Deep Predictive Coding captures contour integration capabilities of the early visual system" (PCOMPBIOL-D-19-01811) for consideration at PLOS Computational Biology. As with all papers peer reviewed by the journal, your manuscript was reviewed by members of the editorial board and by several independent peer reviewers. Based on the reports, we regret to inform you that we will not be pursuing this manuscript for publication at PLOS Computational Biology.

The reviews are attached below this email, and we hope you will find them helpful if you decide to revise the manuscript for submission elsewhere. We are sorry that we cannot be more positive on this occasion. We very much appreciate your wish to present your work in one of PLOS's Open Access publications. 

Thank you for your support, and we hope that you will consider PLOS Computational Biology for other submissions in the future.

Sincerely,

Leyla Isik

Associate Editor

PLOS Computational Biology

Lyle Graham

Deputy Editor

PLOS Computational Biology

Reviewer's Responses to Questions

**Comments to the Authors: **

Reviewer #1: In a series of experiments, the authors examine the role of feedback connections at two levels, “neural” and representational. They do this using a 2 layer Sparse Deep Predictive Coding (SDPC) model varying feedback and sparseness parameters. The model is akin to the V1 and V2 layers of the visual system.

Overall, this seems like a sensible paper with sensible results. It is not clear to what extent its results would be considered surprising or to what extent they advance existing understanding, but it is hard to judge this from my strongly biased perspective in favor of the general nature of the story here. At a computational level, it seems that their model is essentially a backprop auto-encoder with a sparseness constraint, and the idea that an auto-encoder trained on such images learns the features it does, is definitely not novel or surprising. Demonstrating the more detailed effects on noisy images and orientation tuning may be of more interest, but it is essential that the authors consider their results in the context of the large literature on denoising autoencoders. I am not up-to-date on this literature but I would not be surprised if very similar results had been published already.

Thus, I would suggest at least a major revision is in order to better situate this work within the existing computational literature on denoising autoencoders, and related work on how these models compare with known biological data. It may well be that this connection with the biological data has not been clearly established, but given the apparent lack of awareness of a large existing literature, the authors should redouble their efforts to find existing relevant work.

More detailed comments follow.

In the first experiment, they show that adding feedback increases the number of active neurons in the V1 layer and increasing feedback further increases the number of active neurons. The next analysis looks at the effect of feedback on the organization of neurons in the V1 layer. A nice analysis, visually presented and supported by statistics, shows that neurons surrounding what would be the classical receptive field become more aligned with the center neuron’s preferred orientation when there is feedback from the V2 layer. In a further analysis they show that the activity near the center of the interaction map decreases with feedback while the neurons aligned end-to-end with the center field show increased activation. This helps clarify the broad result of increased total activation which is hard to interpret independent of this additional information. The discussion of these results in light of previous literature is minimal and should be specifically discussed in relation to the predictive coding theory as the authors chose this as their base model.

In the second experiment the effect of feedback on degraded images was explored. Images were degraded with Gaussian noise and feedback was varied. The visual representations made by both the first and second layers show clearer images with increased feedback at all levels of degradation. The visual results were supported statistically using a structural similarity measure. But there was mention that the SSIM index showed a slight decrease compared to the original image when a non-degraded stimulus was presented. In light of this result and the progressive improvement with more feedback when images were degraded the authors suggest replacing the scalar controlling feedback strength with a value proportional to the prediction error. But what does it mean in biological terms to weaken or strengthen the feedback? This should be clarified. Though the paper has many references it could do more relating biological mechanisms to computational ones. Is this paper trying to elucidate biological properties of the V1 and V2 layers or suggest how we can improve performance of a predictive coding implementation. The last phrase of the conclusion is “tighten that particular link with neuroscience” but I think the point is not to tighten links but rather to use what we know of neuroscience to improve models and to use models as a tool to understand the computational mechanisms of the human brain.

Minor:

Line 398 - (see Fig. 8 for an illustration) – should this be Figure 2?

Reviewer #2: The authors present a "Sparse Deep Predictive Coding" (SDPC) model which is trained on two datasets and then analyzed in terms of the learned receptive fields, Gestalt continuation, and noisy image reconstruction. While I commend the authors on performing several levels of analysis, I have several concerns:

1. The exact loss function and meaning of k_FB for a two layer network should be clarified. In particular, the use of the k_FB constant as corresponding to "the strength of the feedback connection" is somewhat misleading. With two layers, it seems like total loss function is essentially reconstruction of the inputs (x), reconstruction of the V1 activity map (gamma_1), and the sparsity terms. What k_FB corresponds to then is essentially how much penalization is placed on V1 reconstruction. Couldn't the loss on the inputs (x) also be considered "feedback" to e.g. LGN? At the very least, it should be made explicit what k_FB means and what the total loss function is here.

2. The necessity of sparse coding and/or predictive coding for reproducing the shown effects is not demonstrated. For instance, to demonstrate that sparse coding is necessary, the authors should run the experiments without the L1 loss terms on activity. To demonstrate that predictive coding is necessary, the authors should a) calculate the results from the initial random weights in the network and b) calculate results after training like usual but then randomizing the V2 => V1 feedback weights. Networks with random weights and non-linearities have been shown respond highly to gabor-like stimuli, for instance (Andrew Saxe et al. On random weights and unsupervised feature learning, 2011). Given the ReLU-like non-linearity in the present network, one could imagine that even random "feedback" weights could lead to results like Fig 3. Additionally, as the authors show, without feedback, there is already significantly above chance co-linearity, so it is conceivable again that random feedback weights could slightly boost this (as the trained feedback weights do). 

3. Even with the above caveats, the results supporting the core emphasis on the necessity of feedback are fairly weak. The differences between k_FB=0 and k_FB=1 are relatively small in Fig. 5 and 7 for instance. Also gabor-like receptive fields have been shown in a number of models (sparse coding, predictive coding, supervised FF neural networks, etc.), so this doesn't add any novelty. Additionally, the statement that the model "results in better disambiguation of blurred images at the representational level" is also somewhat misleading. Disambiguation would normally mean that it led to better classification of blurred images for instance, not reconstruction. Such a demonstration, i.e. fitting an SVM on the learned representation and showing that "feedback" helped in a noisy scenario would also be more interesting.

4. The model itself does not seem that novel. Rao & Ballard 1999 already presented a two-layer predictive coding model for image reconstruction that demonstrated Gabor-like receptive fields and other extraclassical receptive field effects. What makes the presented model a significant advancement from the Rao & Ballard model?

Other minor points:

- Line 7: "This feed-forward flow of information is sufficient to account for object categorization in the IT cortical area." => This seems like an overstatement - it can account for some amount of object categorization.

- "the LVis model fitted the psychophysical findings" => use fit instead of fitted

- "To tighten the link with neuroscience, we impose γi to be non-negative and we force Di to have a convolutional structure" => Why is convolution biologically inspired? I agree with similar receptive fields tiled across receptive space, but this phrase should be made more precise. I.e. Convolution corresponds to weight-tying, is there any evidence in biology that this is the case?

- Missing end parenthesis on line 131

- Line 525: "We then explicit the back-projection mechanism used to interpret and visualize inference and learning results" => change to "We then explicitly describe..."

- Line 584: "or" should be "are"

- The training and testing data splits are described. Is there a validation split?

- More explanation of the construction of the interaction map in the main text would be helpful.

- Figure 11 references an "AT&T" database - is this a typo?

- The statement "The baseline represents the co-linearity and co-circularity without feedback" in the figures is confusing - isn't k_FB=0 also "without feedback"?

Overall, while I appreciate the multiple levels of analysis, the lack of convincing evidence of the necessity of either sparse coding or predictive coding (more specifically, the loss on V1 activity map reconstruction) and lack of novelty present concerns.

**Have all data underlying the figures and results presented in the manuscript been provided?**

Reviewer #1: Yes

Reviewer #2: Yes

PLOS authors have the option to publish the peer review history of their article (what does this mean?). If published, this will include your full peer review and any attached files.

Reviewer #1: No

Reviewer #2: No

---

## [Decision Letter · Decision Letter 1]

22 Jul 2020

Dear Mr. Boutin,

Thank you very much for submitting your manuscript "Sparse Deep Predictive Coding captures contour integration capabilities of the early visual system" for consideration at PLOS Computational Biology.

As with all papers reviewed by the journal, your manuscript was reviewed by members of the editorial board and by several independent reviewers. In light of the reviews (below this email), we would like to invite the resubmission of a significantly-revised version that takes into account the reviewers' comments.

I'm very sorry for the long delay -- for reasons that are now shrouded in mystery, I inherited this paper from the previous action editor, and there was massive mis-communication about the reviewers (at least partially because I don't pay attention to emails). Because of that, we essentially ended up with one reviewer. So I read it relatively carefully, and have a few comments in addition to the ones from the reviewer. My comments, in no particular order, follow. The good news, though, that it seems very highly likely that it will be accepted, presumably on the next round.

1. There are way too many initials!!!!! Here's a list, which may or may not be complete,

AE auto-encoder

PC predictive coding

SC sparse coding

CNN convolutional neural network

SDPC Sparse Deep Predictive Coding

cRF classical receptive field

RF Receptive Field

HSC Hierarchical Sparse Codingi

ISTA Iterative Shrinkage Thresholding Algorithms

FISTA not defined, but maybe F=fast?

MAD Median Absolute Deviation

WT Wilcoxon signed-rank test

STL-10 natural images database

CFD face database

SSIM median structural similarity index

This is your paper, and you can write it any way you want, but you should know that all these initials make it insanely hard to read. Especially if you don't read it in one sitting (pretty much the norm). I personally would eliminate every single one of them.

2. You should provide a reference Eq. 1 as soon as it's introduced, since exactly that model appears in ref 67 (although they did not fit it with feedback).

3. Given Eq. 1, it seems that the natural thing to do is minimize a global loss function,

sum_k ||epsilon_k||_2^2 + lambda_k ||gamma_k||_0

or maybe the 1-norm just to make things easier. If I understand things correctly, this corresponds to k_FB=1. Is that correct? If so, it seems worth mentioning that k_FB \\ne 1 would appear to be suboptimal. At least suboptimal relative to the natural cost function.

4. Lines 79-81: "First, the SDPC relies on a biologically plausible unsupervised training through the minimization of local reconstruction errors (in contrast to AEs)." I would disagree with this: forward and backward weights are transposes of each other (something that's not true in the brain), and the weights are convolutional (but see next comment).

5. For the network to be convolutional, don't you have to average the gradients across space? Eq. 7, the weight update, does not seem to do that. Perhaps I missed something?

6. Eq. 4: You don't really mean partial L/partial gamma, since when you compute partial L/partial gamma you ignore the L1 constraint.

7. My suggestion would be to put the the update and learning rules in the main text. Again, it's your paper, so you can write it any way you want. But I found that seeing the update rules makes things easier, not harder. And this is, after all, a computational journal. In fact, I would even put in the derivation, since it's very short.

8. Eq. 6 isn't about stability, it's about fixed points. Correct?

9. Eq. 8: The network does not have access to the back-projection, correct? If so, what is its significance? Is it used only for visualization?

9a. There's a D_0 in Eq. 8. But I don't think D_0 exists. Is that a typo, or did I miss something?

10. I didn't understand Eq. 10 (I believe the reviewer commented on this as well). Consequently, I didn't understand the rest of this section. This is probably the only comment that will take some effort: the "Interaction maps analysis" needs to be understandable, and it currently isn't. At least not by me, and this is sort of my field.

11. Assuming I understand things, and there really is a typo in Eq. 8, then presumably the reconstruction in Figs. 2C and H is D_1^T gamma_1, and in Figs. 2E and J it's D_1^T D_2^T gamma_2? If so, it would be helpful to say so, so that the reader doesn't have to refer to Methods.

12. I agree with the reviewer: when looking at the effects of the feedback at the neural level, it's critical to compare to neural data. If not, it's very hard to make sense of the results. Several of the following comments are along those lines.

13. Stronger feedback means more active neurons. How should we interpret this? Is there any experimental data? Without context, it seems like an isolated fact.

14. Lines 208-12: Inter-stimuli variability goes up with increasing feedback, and is minimum when there is zero feedback. I'm not exactly sure what "inter-stimuli variability" is, but if it's bad, that suggests no feedback is best.

15. Feedback and sparsity improve denoising (usually), but that isn't really the value of these models, right? Presumably the real value is that the variable in the higher layers help with object recognition. If so, this should at pointed out.

16. As a control, you should compare denoising by simply low-pass filtering the noisy image. That would serve as a useful baseline.

We cannot make any decision about publication until we have seen the revised manuscript and your response to the reviewers' comments. Your revised manuscript is also likely to be sent to reviewers for further evaluation.

Sincerely,

Peter E. Latham

Associate Editor

PLOS Computational Biology

Lyle Graham

Deputy Editor

PLOS Computational Biology

I'm very sorry for the long delay -- for reasons that are now shrouded in mystery, I inherited this paper from the previous action editor, and there was massive mis-communication about the reviewers (at least partially because I don't pay attention to emails). Because of that, we essentially ended up with one reviewer. So I read it relatively carefully, and have a few comments in addition to the ones from the reviewer. My comments, in no particular order, follow. The good news, though, that it seems very highly likely that it will be accepted, presumably on the next round.

1. There are way too many initials!!!!! Here's a list, which may or may not be complete,

AE auto-encoder

PC predictive coding

SC sparse coding

CNN convolutional neural network

SDPC Sparse Deep Predictive Coding

cRF classical receptive field

RF Receptive Field

HSC Hierarchical Sparse Codingi

ISTA Iterative Shrinkage Thresholding Algorithms

FISTA not defined, but maybe F=fast?

MAD Median Absolute Deviation

WT Wilcoxon signed-rank test

STL-10 natural images database

CFD face database

SSIM median structural similarity index

This is your paper, and you can write it any way you want, but you should know that all these initials make it insanely hard to read. Especially if you don't read it in one sitting (pretty much the norm). I personally would eliminate every single one of them.

2. You should provide a reference Eq. 1 as soon as it's introduced, since exactly that model appears in ref 67 (although they did not fit it with feedback).

3. Given Eq. 1, it seems that the natural thing to do is minimize a global loss function,

sum_k ||epsilon_k||_2^2 + lambda_k ||gamma_k||_0

or maybe the 1-norm just to make things easier. If I understand things correctly, this corresponds to k_FB=1. Is that correct? If so, it seems worth mentioning that k_FB \\ne 1 would appear to be suboptimal. At least suboptimal relative to the natural cost function.

4. Lines 79-81: "First, the SDPC relies on a biologically plausible unsupervised training through the minimization of local reconstruction errors (in contrast to AEs)." I would disagree with this: forward and backward weights are transposes of each other (something that's not true in the brain), and the weights are convolutional (but see next comment).

5. For the network to be convolutional, don't you have to average the gradients across space? Eq. 7, the weight update, does not seem to do that. Perhaps I missed something?

6. Eq. 4: You don't really mean partial L/partial gamma, since when you compute partial L/partial gamma you ignore the L1 constraint.

7. My suggestion would be to put the the update and learning rules in the main text. Again, it's your paper, so you can write it any way you want. But I found that seeing the update rules makes things easier, not harder. And this is, after all, a computational journal. In fact, I would even put in the derivation, since it's very short.

8. Eq. 6 isn't about stability, it's about fixed points. Correct?

9. Eq. 8: The network does not have access to the back-projection, correct? If so, what is its significance? Is it used only for visualization?

9a. There's a D_0 in Eq. 8. But I don't think D_0 exists. Is that a typo, or did I miss something?

10. I didn't understand Eq. 10 (I believe the reviewer commented on this as well). Consequently, I didn't understand the rest of this section. This is probably the only comment that will take some effort: the "Interaction maps analysis" needs to be understandable, and it currently isn't. At least not by me, and this is sort of my field.

11. Assuming I understand things, and there really is a typo in Eq. 8, then presumably the reconstruction in Figs. 2C and H is D_1^T gamma_1, and in Figs. 2E and J it's D_1^T D_2^T gamma_2? If so, it would be helpful to say so, so that the reader doesn't have to refer to Methods.

12. I agree with the reviewer: when looking at the effects of the feedback at the neural level, it's critical to compare to neural data. If not, it's very hard to make sense of the results. Several of the following comments are along those lines.

13. Stronger feedback means more active neurons. How should we interpret this? Is there any experimental data? Without context, it seems like an isolated fact.

14. Lines 208-12: Inter-stimuli variability goes up with increasing feedback, and is minimum when there is zero feedback. I'm not exactly sure what "inter-stimuli variability" is, but if it's bad, that suggests no feedback is best.

15. Feedback and sparsity improve denoising (usually), but that isn't really the value of these models, right? Presumably the real value is that the variable in the higher layers help with object recognition. If so, this should at pointed out.

16. As a control, you should compare denoising by simply low-pass filtering the noisy image. That would serve as a useful baseline.

Reviewer's Responses to Questions

**Comments to the Authors:**

Reviewer #3: Let me first clarify, that I was not a reviewer of the previous version of the manuscript. My expertise is in predictive coding networks, but no so much in the visual cortex.

This manuscript scales up the original predictive coding model of Rao & Ballard (which was published in 1999 and hence simulated relatively small network) to a much larger size, and trains it on large number of realistic images. The manuscript then analyses the contour integration properties of this network. The paper is relatively clearly written, and I only have several minor comments.

Comments:

Is it possible to qualitatively compare the model with experimental data? Are there any experimental studies that allow extracting quantities similar to those in Fig 4-7? If yes, it would be ideal to add data points to these graphs that would correspond to experiments. If not, please at least relate the patterns of data in these figures to specific experimental results.

Intro: It is unclear what the biological question the paper addresses. What are the phenomena that have not been explained by the previous models, and this paper addresses? Please state this explicitly.

Discussion: Does the model make any experimentally testable predictions that could be tested in neurophysiological experiments? If so, it would be worth to state them.

Line 82: It is unclear what “FISTA” stands for – please clarify.

Line 95: Symbol “j” does not appear until Equation 10 (at the very end of the paper), so I suggest to introduce it there rather than at the start of the paper.

Line 103: “l_0 pseudo-norm” – I do not understand what it means – please clarify.

Line 452: “an Hebbian” -> “a Hebbian”

Table 1: I do not understand what “number of features, number of channels” are. Please clarify.

Eq 10: I do not understand the notation in the equation. What is the meaning of the symbol with three lines? Is it “define”? Is so, it does not make sense to use on the left hand side a symbol (gamma_1) that was already used, and please use a different symbol. What does “{…}” mean – is it an average or is it a set?

Eq 14: What is “atan2”? Are there some brackets missing?

**Have all data underlying the figures and results presented in the manuscript been provided?**

Reviewer #3: Yes

PLOS authors have the option to publish the peer review history of their article (what does this mean?). If published, this will include your full peer review and any attached files.

Reviewer #3: No
---

## [Decision Letter · Decision Letter 2]

8 Nov 2020

Dear Mr. Boutin,

Thank you very much for submitting your manuscript "Sparse Deep Predictive Coding captures contour integration capabilities of the early visual system" for consideration at PLOS Computational Biology. As with all papers reviewed by the journal, your manuscript was reviewed by members of the editorial board and by several independent reviewers. The reviewers appreciated the attention to an important topic. Based on the reviews, we are likely to accept this manuscript for publication, providing that you modify the manuscript according to the review recommendations.

As you can see, the reviewer was pretty happy with the paper. However, I still found the paper very confusing in some places -- to the point where I got lost. Most are easily fixable, but they should be fixed.

I will admit, I was a bit torn. After all, it's your paper, and you should be free to write it any way you want, so long as it isn't wrong. However, given that this is more or less my field, if I'm confused, probably other people will be too. So this is my attempt to save people some time.

Feel free to contact me (at Peter Latham <pel@gatsby.ucl.ac.uk>) if you have any questions, or you want me to look at changes you have made, before you resubmit. That will be a _lot_ more efficient than the standard review process. And since the paper is effectively in, that seems OK to me.

Specific comments:

1. Please put the figures in the text, where they belong. I know, PLoS says not to, but ignore them. It's very hard to read when I have to keep flipping to the end. And it's especially hard when the figures and figure captions are separated.

2. Line 123-4, "We call D_i^eff the back-projection of D_i into the visual space (see Fig. S7)." D_i^eff needs to be defined here; otherwise, the sentence doesn't add much.

3. I still don't understand why the gradients are convolutional. To my simple-minded way of thinking, if the gradients are convolutional, the weights will end up with the form

   w_{ij} = w_{i-j}

where i and j index position, with j referring to the layer below i. I don't see anything in your equations to suggest that that's what you're doing. That should be fixed.

Note that it's possible that we have different definitions of convolution, and mine could easily be wrong.

4. Interaction map needs to be explained better, in the main text, given how important they are. Specific comments follow, indicating may confusion.

a. In the section starting on line 244, equations are needed. I simply could not figure out what you were talking about. See next several comments.

b. Lines 736-8: "Let’s first remind that our V1 representation (denoted gamma_1) is a 3-dimensional tensor in which the first dimension is describing the feature space (denoted theta), and the 2 last dimensions are related to the spatial position (x and y respectively)."

This is the first I've heard of this! Did I missed it? If so, please tell me where it is; if not, it should be explained early on. It seems very important.

c. Eq. 10: "(see Eq. 10)" should not go right before Eq. 10. But that's a detail; the main problem is that I couldn't understand Eq. 10. An equal sign would help.

d. Lines 746-7: "In practice, we estimate theta by fitting the first layer Receptive Fields (RFs) with Gabor filters [70]."

This doesn't make sense to me. Which isn't a surprise, since I don't know what Eq. 10 means.

e. I sort of understand steps 1-3, although lines 754-7,

"In practice, we consider the positions of the 10 neurons showing the strongest activity. Second, we extract a spatial neighborhood of size 9x9 centered on these strongly responsive neurons."

are confusing. My first thought was: "How can a single neighborhood be centered on all 10 neurons?" Only at the end do you say that you carry out the process for the 10 neurons, and average. Please make this clear earlier; I spent a lot of time trying to figure out what was going on.

Sincerely,

Peter E. Latham

Associate Editor

PLOS Computational Biology

Lyle Graham

Deputy Editor

PLOS Computational Biology

[LINK]

Reviewer's Responses to Questions

**Comments to the Authors:**

Reviewer #3: The Authors fully addressed my comments. I only have two tiny suggestions:

Line 93: Since my previous review, a new preprint has been published which also extended predictive coding to convolutional setting:

Millidge, B., Tschantz, A., & Buckley, C. L. (2020). Predictive Coding Approximates Backprop along Arbitrary Computation Graphs. arXiv preprint arXiv:2006.04182.

Hence the description of novel contributions of Author’s manuscript needs to be adjusted, and the Author may wish to discuss the relationship of their work to that paper.

Lines 117-118: In response to my comment, the Authors added an explanation for why l0 is a pseudo-norm. However, I feel it would rather be more helpful to instead provide the definition of l0, for example as they wrote in the response letter: “the l0-norm is just counting the number of strictly positive scalars, and does not depend on their amplitude”.

**Have all data underlying the figures and results presented in the manuscript been provided?**

Reviewer #3: Yes

PLOS authors have the option to publish the peer review history of their article (what does this mean?). If published, this will include your full peer review and any attached files.

Reviewer #3: No
---

## [Editor Report · Decision Letter 3]

12 Dec 2020

Dear Mr. Boutin,

We are pleased to inform you that your manuscript 'Sparse Deep Predictive Coding captures contour integration capabilities of the early visual system' has been provisionally accepted for publication in PLOS Computational Biology.

Best regards,

Peter E. Latham

Associate Editor

PLOS Computational Biology

Lyle Graham

Deputy Editor

PLOS Computational Biology

---

## [Editor Report · Acceptance letter]

21 Jan 2021

PCOMPBIOL-D-19-01811R3 

Sparse Deep Predictive Coding captures contour integration capabilities of the early visual system

Dear Dr Boutin,

I am pleased to inform you that your manuscript has been formally accepted for publication in PLOS Computational Biology. Your manuscript is now with our production department and you will be notified of the publication date in due course.

With kind regards,

Alice Ellingham
